# Approximate Cross-Validation for Structured Models

**Soumya Ghosh**[*][†]
MIT-IBM Watson AI Lab
IBM Research
ghoshso@us.ibm.com

**William T. Stephenson**[*]
MIT CSAIL
MIT-IBM Watson AI Lab
wtstephe@mit.edu

**Tin D. Nguyen**
MIT CSAIL
MIT-IBM Watson AI Lab
tdn@mit.edu

**Sameer K. Deshpande**
MIT CSAIL
MIT-IBM Watson AI Lab
sameerd@alum.mit.edu

**Tamara Broderick**
MIT CSAIL
MIT-IBM Watson AI Lab
tbroderick@csail.mit.edu

## Abstract

Many modern data analyses benefit from explicitly modeling dependence structure in data – such as measurements across time or space, ordered words in a sentence, or genes in a genome. A gold standard evaluation technique is structured cross-validation (CV), which leaves out some data subset (such as data within a time interval or data in a geographic region) in each fold. But CV here can be prohibitively slow due to the need to re-run already-expensive learning algorithms many times. Previous work has shown approximate cross-validation (ACV) methods provide a fast and provably accurate alternative in the setting of empirical risk minimization. But this existing ACV work is restricted to simpler models by the assumptions that (i) data across CV folds are independent and (ii) an exact initial model fit is available. In structured data analyses, both these assumptions are often untrue. In the present work, we address (i) by extending ACV to CV schemes with dependence structure between the folds. To address (ii), we verify – both theoretically and empirically – that ACV quality deteriorates smoothly with noise in the initial fit. We demonstrate the accuracy and computational benefits of our proposed methods on a diverse set of real-world applications.

## 1 Introduction

Models with complex dependency structures have become standard machine learning tools in analyses of data from science, social science, and engineering fields. These models are used to characterize disease progression [Sukkar et al., 2012, Wang et al., 2014, Sun et al., 2019], to track crime in a city [Balocchi and Jensen, 2019, Balocchi et al., 2019], and to monitor and potentially manage traffic flow [Ihler et al., 2006, Zheng and Liu, 2017] among many other applications. The potential societal impact of these methods necessitates that they be used and evaluated with care. Indeed, recent work [Musgrave et al., 2020] has emphasized that hyperparameter tuning and assessment with *cross-validation* (CV) [Stone, 1974, Geisser, 1975] is crucial to trustworthy and meaningful analysis of modern, complex machine learning methods.

While CV offers a conceptually simple and widely used tool for evaluation, it can be computationally prohibitive in complex models. These models often already face severe computational demands to fit just once, and CV requires multiple re-fits. To address this cost, recent authors [Beirami et al., 2017, Rad and Maleki, 2020, Giordano et al., 2019] have proposed *approximate* CV (ACV) methods; their

---

[*]Joint first authorship
[†]Also with the Center for Computational Health

work demonstrates that ACV methods perform well in both theory and practice for a collection of practical models. These methods take two principal forms: one approximation based on a Newton step (NS) [Beirami et al., 2017, Rad and Maleki, 2020] and one based on the classical infinitesimal jackknife (IJ) from statistics [Koh and Liang, 2017, Beirami et al., 2017, Giordano et al., 2019]. Though both ACV forms show promise, there remain major roadblocks to applying either NS or IJ to models with dependency structure. First, all existing ACV theory and algorithms assume that data dropped out by each CV fold are independent of the data in the other folds. But to evaluate time series models, for instance, we often drop out data points in various segments of time. Or we might drop out data within a geographic region to evaluate a spatiotemporal model. In all of these cases, the independence assumption would not apply. Second, NS methods require recomputation and inversion of a model's Hessian matrix at each CV fold. In the complex models we consider here, this cost can itself be prohibitive. Finally, existing theory for IJ methods requires an exact initial fit of the model – and authors so far have taken great care to obtain such a fit [Giordano et al., 2019, Stephenson and Broderick, 2020]. But practitioners learning in e.g. large sequences or graphs typically settle for an approximate fit to limit computational cost.

In this paper, we address these concerns and thereby expand the reach of ACV to include more sophisticated models with dependencies among data points and for which exact model fits are infeasible. To avoid the cost of matrix recomputation and inversion across folds, we here focus on the IJ, rather than the NS. In particular, in Section 3, we develop IJ approximations for dropping out individual nodes in a dependence graph. Our methods allow us e.g. to leave out points within, or at the end of, a time series – but our methods also apply to more general Markov random fields, without a strict chain structure. In Section 4, we demonstrate that the IJ yields a useful ACV method even without an exact initial model fit. In fact, we show that the quality of the IJ approximation decays with the quality of the initial fit in a smooth and interpretable manner. Finally, we demonstrate our method on a diverse set of real-world applications and models in Section 5 and Appendix M. These include count data analysis with time-varying Poisson processes, named entity recognition with neural conditional random fields, motion capture analysis with auto-regressive hidden Markov models, and a spatial analysis of crime data with hidden Markov random fields.

## 2  Structured models and cross-validation

### 2.1  Structured models

Throughout we consider two types of models: (1) hidden Markov random fields (MRFs) with observations $\mathbf{x}$ and latent variables $\mathbf{z}$ and (2) conditional random fields (CRFs) with inputs (i.e., covariates) $\mathbf{x}$ and labels $\mathbf{z}$, both observed. Our developments for hidden MRFs and CRFs are very similar, but with slight differences. We detail MRFs in the main text; throughout, we will refer the reader to the appendix for the CRF treatment. We first give an illustrative example of MRFs and then the general formulation; a CRF overview appears in Appendix F.

**Example**: Hidden Markov Models (HMMs) capture sequences of observations such as words in a sentence or longitudinally measured physiological signals. Consider an HMM with $N$ (independent) sequences, $T$ time steps, and $K$ states. We take each observation to have dimension $R$. So the $t$th observed element in the $n$th sequence is $x_{nt} \in \mathbb{R}^R$, and the latent $z_{nt} \in [K] := \{1, \ldots, K\}$. The model is specified by (1) a distribution on the initial latent state $p(z_{n1}) = \mathrm{Cat}(z_{n1} \mid \pi)$, where Cat is the categorical distribution and $\pi \in \Delta_{K-1}$, the $K-1$ simplex; (2) a $K \times K$ transition matrix $A$ with columns $A_k \in \Delta_{K-1}$ and $p(z_{nt} \mid z_{n,t-1}) = \mathrm{Cat}(z_{nt} \mid A_{z_{n,t-1}})$; and (3) emission distributions $F$ with parameters $\theta_k$ such that $p(x_{nt} \mid z_{nt}) = F(x_{nt} \mid \theta_{z_{nt}})$. We collect all parameters of the model in $\Theta := \{\pi, \{A_k\}_{k=1}^K, \{\theta_k\}_{k=1}^K\}$. We consider $\Theta$ as a vector of length $D$. We may have a prior $p(\Theta)$.

More generally, we consider **(hidden) MRFs** with $N$ structured observations $\mathbf{x}_n$ and latents $\mathbf{z}_n$, independent across $n \in [N]$. We index single observations of dimension $R$ (respectively, latents) within the structure by $t \in [T]$: $x_{nt} \in \mathbb{R}^R$ (respectively, $z_{nt}$). Our experiments will focus on bounded, discrete $z_{nt}$ (i.e., $z_{nt} \in [K]$), but we use more inclusive notation (that might e.g. apply to continuous latents) when possible. We consider models with parameters $\Theta \in \mathbb{R}^D$ and a single emission factor for each latent.

$$-\log p(\mathbf{x}, \mathbf{z}; \Theta) = Z(\Theta) + \sum_{n=1}^N \left\{ \left[ \sum_{t \in [T]} \psi_t(x_{nt}, z_{nt}; \Theta) \right] + \left[ \sum_{\mathbf{c} \in \mathcal{F}} \phi_{\mathbf{c}}(z_{n\mathbf{c}}; \Theta) \right] \right\}, \quad (1)$$

where $z_{n\mathbf{c}} := (z_{nt})_{t \in \mathbf{c}}$ for $\mathbf{c} \subseteq [T]$; $\psi_t$ is a log factor mapping $(x_{nt}, z_{nt})$ to $\mathbb{R}$; $\phi_{\mathbf{c}}$ is a log factor mapping collections of latents, indexed by $\mathbf{c}$, to $\mathbb{R}$; $\mathcal{F}$ collects the subsets indexing factors; and $Z(\Theta)$ is a negative log normalizing constant. HMMs, as described above, are a special case; see Appendix B for details. For any MRF, we can learn the parameters by marginalizing the latents and maximizing the posterior, or equivalently the joint, in $\Theta$. Maximum likelihood estimation is the special case with formal prior $p(\Theta)$ constant across $\Theta$.

$$\hat{\Theta} := \operatorname*{argmin}_{\Theta} \ - \log p(\mathbf{x}; \Theta) - \log p(\Theta) = \operatorname*{argmin}_{\Theta} \ - \log \int_{\mathbf{z}} p(\mathbf{x}, \mathbf{z}; \Theta) \, d\mathbf{z} - \log p(\Theta). \quad (2)$$

## 2.2 Challenges of cross-validation and approximate cross-validation in structured models

In CV procedures, we iteratively leave out some data in order to diagnose variation in $\hat{\Theta}$ under natural data variability or to estimate the predictive accuracy of our model. We consider two types of CV of interest in structured models; we make these formulations precise later. (1) We say that we consider *leave-within-structure-out* CV (LWCV) when we remove some data points $x_{nt}$ within a structure and learn on the remaining data points. For instance, we might try to predict crime in certain census tracts based on observations in other tracts. Often in this case $N = 1$ [Celeux and Durand, 2008], [Hyndman and Athanasopoulos, 2018, Chapter 3.4], and we assume LWCV has $N = 1$ for notational simplicity in what follows. (2) We say that we consider *leave-structure-out* CV (LSCV) when we leave out entire $\mathbf{x}_n$ for either a single $n$ or a collection of $n$. For instance, with a state-space model of gene expression, we might predict some individuals' gene expression profiles given other individuals' profiles. In this case, $N \gg 1$ [Rangel et al., 2004, DeCaprio et al., 2007]. In either (1) or (2), the goal of CV is to consider multiple folds, or subsets of data, left out to assess variability and improve estimation of held-out error. But every fold incurs the cost of the learning procedure in Eq. (2). Indeed, practitioners have explicitly noted the high cost of using multiple folds and have resorted to using only a few, large folds [Celeux and Durand, 2008], leading to biased or noisy estimates of the out-of-sample variability.

A number of researchers have addressed the prohibitive cost of CV with approximate CV (ACV) procedures for simpler models [Beirami et al., 2017, Rad and Maleki, 2020, Giordano et al., 2019]. Existing work focuses on the following learning problem with weights $\mathbf{w} \in \mathbb{R}^J$:

$$\hat{\Theta}(\mathbf{w}) = \operatorname*{argmin}_{\Theta} \ \sum_{j \in [J]} w_j f_j(\Theta) + \lambda R(\Theta), \quad (3)$$

where $\forall j \in [J], f_j, R : \mathbb{R}^D \to \mathbb{R}$ and $\lambda \in \mathbb{R}_+$. When the weight vector $\mathbf{w}$ equals the all-ones vector $\mathbf{1}_J$, we recover a regularized empirical loss minimization problem. By considering all weight vectors with one weight equal to zero, we recover the folds of leave-one-out CV; other forms of CV can be similarly recovered. The notation $\hat{\Theta}(\mathbf{w})$ emphasizes that the learned parameter values depend on the weights.

To see if this framework applies to LWCV or LSCV, we can interpret $f_j$ as a negative log likelihood (up to normalization) for the $j$th data point and $\lambda R$ as a negative log prior. Then the likelihood corresponding to the objective of Eq. (3) factorizes as $p(\mathbf{x} \mid \Theta) = \prod_{j \in J} p(x_j \mid \Theta) \propto \prod_{j \in J} \exp(-f_t(\Theta))$. This factorization amounts to an independence assumption across the $\{x_j\}_{j \in [J]}$. In the case of LWCV, with $N = 1$, $j$ must serve the role of $t$, and $J = T$. But the $x_t$ are not independent, so we cannot apply existing ACV methods. In the LSCV case, $N \geq 1$, and $j$ in Eq. (3) can be seen as serving the role of $n$, with $J = N$. Since the $\mathbf{x}_n$ are independent, Eq. (3) can express LSCV folds.

**Previous ACV work** provides two primary options for the LSCV case. We give a brief review here, but see Appendix A for a more detailed review. One option is based on taking a single Newton step on the LSCV objective starting from $\hat{\Theta}(\mathbf{1}_T)$ [Beirami et al., 2017, Rad and Maleki, 2020]. Except in special cases – such as leave-one-out CV for generalized linear models – this Newton-step approach requires both computing and inverting a new Hessian matrix for each fold, often a prohibitive expense; see Appendix H for a discussion. An alternative method [Koh and Liang, 2017, Beirami et al., 2017, Giordano et al., 2019] based on the *infinitesimal jackknife* (IJ) from statistics [Jaeckel, 1972, Efron, 1981] constructs a Taylor expansion of $\hat{\Theta}(\mathbf{w})$ around $\mathbf{w} = \mathbf{1}_T$. For any model of the form in Eq. (3), the IJ requires just a single Hessian matrix computation and inversion. Therefore, we focus on the IJ for LSCV and use the IJ for inspiration when developing LWCV below. However, all existing IJ theory and empirics require access to an *exact* minimum for $\hat{\Theta}(\mathbf{1}_J)$. Indeed, previous

authors [Giordano et al., 2019, Stephenson and Broderick, 2020] have taken great care to find an exact minimum of Eq. (3). Unfortunately, for most complex, structured models with large datasets, finding an exact minimum requires an impractical amount of computation. Others [Bürkner et al., 2020] have developed ACV methods for Bayesian time series models and for Bayesian models without dependence structures [Vehtari et al., 2017]. Our development here focuses on empirical risk minimization and is not restricted to temporal models.

In the following, we extend the reach of ACV beyond LSCV and address the issue of inexact optimization. In Section 3, we adapt the IJ framework to the LWCV problem for structured models. In Section 4, we show theoretically that both our new IJ approximation for LWCV and the existing IJ approximation applied to LSCV are not overly dependent on having an exact optimum. We support both of these results with practical experiments in Section 5.

## 3   Cross-validation and approximate cross-validation in structured models

We first specify a weighting scheme, analogous to Eq. (3), to describe LWCV in structured models; then we develop an ACV method using this scheme. Recall that CV in independent models takes various forms such as leave-$k$-out and $k$-fold CV. Similarly, we consider the possibility of leaving out[3] multiple arbitrary sets of data indices $\mathbf{o} \in \mathcal{O}$, where each $\mathbf{o} \subseteq [T]$. We have two options for how to leave data out in hidden MRFs; see Appendix G for CRFs. (A) For each data index $t$ left out, we leave out the data point $x_t$ but we retain the latent $z_t$. For instance, in a time series, if data is missing in the middle of the series, we still know the time relation between the surrounding points, and would leave in the latent to maintain this relation. (B) For each data index $t$ left out, we leave out the data point $x_t$ *and* the latent $z_t$. For instance, consider data in the future of a time series or pixels beyond the edge of a picture. We typically would not include the possibility of all possible adjacent latents in such a structure, so leaving out $z_t$ as well is more natural. In either case, analogous to Eq. (3), $\hat{\Theta}(\mathbf{w})$ is a function of $\mathbf{w}$ computed by minimizing the negative log joint $-\log p(\mathbf{x}; \Theta, \mathbf{w}) - \log p(\Theta)$, now with $\mathbf{w}$ dependence, in $\Theta$. For case (A), we adapt Eq. (1) (with $N = 1$) and Eq. (2) with a weight $w_t$ for each $x_t$ term:

$$\hat{\Theta}(\mathbf{w}) = \underset{\Theta}{\operatorname{argmin}}\ Z(\Theta, \mathbf{w}) + \int_{\mathbf{z}} \left[ \sum_{t \in [T]} w_t \psi_t(x_t, z_t; \Theta) \right] + \left[ \sum_{\mathbf{c} \in \mathcal{F}} \phi_{\mathbf{c}}(z_{\mathbf{c}}; \Theta) \right]\ d\mathbf{z} - \log p(\Theta). \quad (4)$$

Note that the negative log normalizing constant $Z(\Theta, \mathbf{w})$ may now depend on $\mathbf{w}$ as well. For case (B), we adapt Eq. (1) and Eq. (2) with a weight $w_t$ for each term with $x_t$ or $z_t$:

$$\hat{\Theta}(\mathbf{w}) = \underset{\Theta}{\operatorname{argmin}}\ Z(\Theta, \mathbf{w}) + \int_{\mathbf{z}} \left[ \sum_{t \in [T]} w_t \psi_t(x_t, z_t; \Theta) \right] + \left[ \sum_{\mathbf{c} \in \mathcal{F}} \left( \prod_{t \in \mathbf{c}} w_t \right) \phi_{\mathbf{c}}(z_{\mathbf{c}}; \Theta) \right]\ d\mathbf{z} - \log p(\Theta). \quad (5)$$

In both cases, the choice $\mathbf{w} = \mathbf{1}_T$ recovers the original learning problem. Likewise, setting $\mathbf{w} = \mathbf{w_o}$, where $\mathbf{w_o}$ is a vector of ones with $w_t = 0$ if $t \in \mathbf{o}$, drops out the data points in $\mathbf{o}$ (and latents in case (B)). We show in Appendix E that these two schemes are equivalent in the case of chain-structured graphs when $\mathbf{o} = \{T', T' + 1, \ldots, T\}$ but also that they are not equivalent in general. We thus consider both schemes going forward.

The expressions above allow a unifying viewpoint on LWCV but still require re-solving $\hat{\Theta}(\mathbf{w_o})$ for each new CV fold $\mathbf{o}$. To avoid this expense, we propose to use an IJ approach. In particular, as discussed by Giordano et al. [2019], the intuition of the IJ is to notice that, subject to regularity conditions, a small change in $\mathbf{w}$ induces a small change in $\hat{\Theta}(\mathbf{w})$. So we propose to approximate $\hat{\Theta}(\mathbf{w_o})$ with $\hat{\Theta}_{\mathrm{IJ}}(\mathbf{w_o})$, a first-order Taylor series expansion of $\hat{\Theta}(\mathbf{w})$ as a function of $\mathbf{w}$ around $\mathbf{w} = \mathbf{1}_T$. We follow Giordano et al. [2019] to derive this expansion in Appendix J. Our IJ based approximation is applicable when the following conditions hold,

**Assumption 1.** *The model is fit via optimization (e.g. MAP or MLE).*

**Assumption 2.** *The model objective is twice differentiable and the Hessian matrix is invertible at the initial model fit $\hat{\Theta}$.*

**Assumption 3.** *The model fits across CV folds, $\hat{\Theta}^{\backslash \mathbf{o}}$, can be written as optima of the same weighted objective for all folds $\mathbf{o}$ (e.g. as in Eqs. (4) and (5)).*

We summarize our method and define $\hat{\Theta}_{\mathrm{ACV}}$, with three arguments, in Algorithm 1; we define $\hat{\Theta}_{\mathrm{IJ}}(\mathbf{w_o}) := \hat{\Theta}_{\mathrm{ACV}}(\hat{\Theta}(\mathbf{1}_T), \mathbf{x}, \mathbf{o})$.

---

**Algorithm 1** Approximate leave-within-structure-out cross-validation for all folds $\mathbf{o} \in \mathcal{O}$

---

**Require:** $\Theta_1, \mathbf{x}, \mathcal{O}$
1: Define *weighted* marginalization over $\mathbf{z}$: $\log p(\mathbf{x}; \Theta, \mathbf{w}) = \textsc{WeightedMarg}(\mathbf{x}, \Theta, \mathbf{w})$.

2: Compute $H = \left. \frac{\partial^2 \log p(\mathbf{x}; \Theta, \mathbf{w}) + \log p(\Theta)}{\partial \Theta \partial \Theta^\top} \right|_{\Theta = \Theta_1, \mathbf{w} = \mathbf{1}_T}$

3: Compute matrix $J = (J_{dt}) := \left( \left. \frac{\partial^2 \log p(\mathbf{x}; \Theta, \mathbf{w}) + \log p(\Theta)}{\partial \Theta_d \partial w_t} \right|_{\Theta = \Theta_1, \mathbf{w} = \mathbf{1}_T} \right)$

4: **for** $\mathbf{o} \in \mathcal{O}$, **do:** $\hat{\Theta}_{\mathrm{ACV}}(\Theta_1, \mathbf{x}, \mathbf{o}) := \Theta_1 + \sum_{t \in \mathbf{o}} H^{-1} J_t$      # $J_t$ is the $t$th column of $J$

5: **return** $\{\hat{\Theta}_{\mathrm{ACV}}(\Theta_1, \mathbf{x}, \mathbf{o})\}_{\mathbf{o} \in \mathcal{O}}$

---

First, note that the proposed procedure applies to either weighting style (A) or (B) above; they each determine a different $\log p(\mathbf{x}, \Theta; \mathbf{w})$ in Algorithm 1. We provide analogous LSCV algorithms for MRFs and CRFs in Algorithms 2 and 3 (Appendices C and G). Next, we compare the cost of our proposed ACV methods to exact CV. In what follows, we consider the initial learning problem $\hat{\Theta}(\mathbf{1}_T)$ a fixed cost and focus on runtime after that computation. We consider running CV for all folds $\mathbf{o} \in \mathcal{O}$ in the typical case where the number of data points left out of each fold, $|\mathbf{o}|$, is constant.

**Proposition 1.** *Let $M$ be the cost of a marginalization, i.e., running* WEIGHTEDMARG*; let $N \geq 1$ be the number of independent structures; and let $S$ be the maximum number of steps used to fit the parameter in our optimization procedure. The cost of any one of our ACV algorithms (Algorithms 1, 2 and 3) is in $O(MN + D^3 + D^2 |\mathbf{o}| |\mathcal{O}|)$. Exact CV is in $O(MNS|\mathcal{O}|)$.*

*Proof.* For each of the $|\mathcal{O}|$ folds of CV and each of the $N$ structures, we compute the marginalization (cost $M$) at each of the $S$ steps of the optimization procedure. In our ACV algorithms, we compute $H$ and $J$ with automatic differentiation tools [Baydin et al., 2018]. The results of Bartholomew-Biggs et al. [2000] demonstrate that $H$ and $J$ each require the same computation (up to a constant) as WEIGHTEDMARG. So, across $N$, we incur cost $MN$. We then incur a $O(D^3)$ cost to invert[4] $H$. The remaining cost is from the for loop. $\square$

In structured problems, we generally expect $M$ to be large; see Appendix D for a discussion of the costs, including in the special case of chain-structured MRFs and CRFs. And for reliable CV, we want $|\mathcal{O}|$ to be large. So we see that our ACV algorithms reap a savings by, roughly, breaking up the product of these terms into a sum and avoiding the further $S$ multiplier.

## 4 IJ behavior under inexact optimization

By envisioning the IJ as a Taylor series approximation around $\hat{\Theta}(\mathbf{1}_T)$, the approximations for LWCV (Algorithm 1) and LSCV (Algorithms 2 and 3 in the appendix) assume we have access to the exact optimum $\hat{\Theta}(\mathbf{1}_T)$. In practice, though, especially in complex problems, computational considerations often require using an inexact optimum. More precisely, any optimization algorithm returns a sequence of parameter values $(\Theta^{(s)})_{s=1}^S$. Ideally the values $\Theta^{(S)}$ will approach the optimum $\hat{\Theta}(\mathbf{1}_T)$ as $S \to \infty$. But we often choose $S$ such that $\Theta^{(S)}$ is much farther from $\hat{\Theta}(\mathbf{1}_T)$ than machine precision. In practice, then, we input $\Theta^{(S)}$ (rather than $\hat{\Theta}(\mathbf{1}_T)$) to Algorithm 1. We now check that the error induced by this substitution is acceptably low.

We focus here on a particular use of CV: estimating out-of-sample loss. For simplicity, we discuss the $N = 1$ case here; see Appendix K for the very similar $N \geq 1$ case. For each fold $\mathbf{o} \in \mathcal{O}$,

we compute $\hat{\Theta}(\mathbf{w_o})$ from the points kept in and then calculate the loss (in our experiments here, negative log likelihood) on the left-out points. I.e. the CV estimate of the out-of-sample loss is $\mathcal{L}_{\text{CV}} := (1/|\mathcal{O}|)\sum_{\mathbf{o}\in\mathcal{O}} -\log p(x_\mathbf{o} \mid x_{[T]-\mathbf{o}}; \hat{\Theta}(\mathbf{w_o}))$, where $-\log p$ may come from either weighting scheme (A) or (B). See Appendix K for an extension to CV computed with a generic loss $\ell$. We approximate $\mathcal{L}_{\text{CV}}$ using some $\Theta$ as input to Algorithm 1; we denote this approximation by $\mathcal{L}_{\text{IJ}}(\Theta) := (1/|\mathcal{O}|)\sum_{\mathbf{o}\in\mathcal{O}} -\log p(x_\mathbf{o} \mid x_{[T]-\mathbf{o}}; \hat{\Theta}_{\text{ACV}}(\Theta, \mathbf{x}, \mathbf{o}))$.

Below, we will bound the error in our approximation: $|\mathcal{L}_{\text{CV}} - \mathcal{L}_{\text{IJ}}(\Theta^{(S)})|$. There are two sources of error. (1) The difference in loss between exact CV and the exact IJ approximation, $\varepsilon_{\text{IJ}}$ in Eq. (6). (2) The difference in the parameter value, $\varepsilon_\Theta$ in Eq. (6), which will control the difference between $\mathcal{L}_{\text{IJ}}(\hat{\Theta}(\mathbf{1}_T))$ and $\mathcal{L}_{\text{IJ}}(\Theta^{(S)})$.

$$\varepsilon_{\text{IJ}} := |\mathcal{L}_{\text{CV}} - \mathcal{L}_{\text{IJ}}(\hat{\Theta}(\mathbf{1}_T))|, \quad \varepsilon_\Theta := \|\Theta^{(S)} - \hat{\Theta}(\mathbf{1}_T)\|_2 \tag{6}$$

Our bound below will depend on these constants. We observe that empirics, as well as theory based on the Taylor series expansion underlying the IJ, have established that $\varepsilon_{\text{IJ}}$ is small in various models; we expect the same to hold here. Also, $\varepsilon_\Theta$ should be small for large enough $S$ according to the guarantees of standard optimization algorithms. We now state some additional regularity assumptions before our main result.

**Assumption 4.** *Take any ball $B \subset \mathbb{R}^D$ centered on $\hat{\Theta}(\mathbf{1}_T)$ and containing $\Theta^{(S)}$. We assume the objective $-\log p(\mathbf{x}; \Theta, \mathbf{1}_T) - p(\Theta)$ is strongly convex with parameter $\lambda_{\min}$ on $B$. Additionally, on $B$, we assume the derivatives $g_t(\Theta) := \partial^2 \log p(\mathbf{x}; \Theta, \mathbf{w})/\partial\Theta\partial w_t$ are Lipschitz continuous with constant $L_g$ for all $t$, and the inverse Hessian of the objective is Lipschitz with parameter $L_{Hinv}$. Finally, on $B$, take $\log p(\mathbf{x}; \Theta, \mathbf{w_o})$ to be a Lipschitz function of $\Theta$ with parameter $L_p$ for all $\mathbf{w_o}$.*

We make a few remarks on the restrictiveness of these assumptions. First, while few structured models have objectives that are even convex (e.g., the label switching problem for HMMs guarantees non-convexity), we expect most objectives to be locally convex around an exact minimum $\hat{\Theta}(\mathbf{1}_T)$; Assumption 4 requires that the objective in fact be strongly locally convex. Next, while the Lipschitz assumption on the $g_t$ may be hard to interpret in general, we note that it takes on a particularly simple form in the setup of Eq. (3), where we have $g_t = \nabla f_t$. Finally, we note that the condition that the inverse Hessian is Lipschitz is not much of an additional restriction. E.g., if $\nabla p(\Theta)$ is also Lipschitz continuous, then the entire objective has a Lipschitz gradient, and so its Hessian is bounded. As it is also bounded below by strong convexity, we find that the inverse Hessian is bounded above and below, and thus is Lipschitz continuous. We now state our main result.

**Proposition 2.** *The approximation error of $\mathcal{L}_{\text{IJ}}(\Theta^{(S)})$ satisfies the following bound:*

$$|\mathcal{L}_{\text{IJ}}(\Theta^{(S)}) - \mathcal{L}_{\text{CV}}| \leq C\varepsilon_\theta + \varepsilon_{\text{IJ}}, \tag{7}$$

$$where \quad C := L_p + \frac{L_p L_g}{\lambda_{\min}} + \frac{L_p L_{Hinv}}{|\mathcal{O}|} \sum_{\mathbf{o}\in\mathcal{O}} \left\| \sum_{t\in\mathbf{o}} \nabla g_t(\hat{\Theta}(\mathbf{1}_T)) \right\|_2 .$$

See Appendix K for a proof. Note that, while $C$ may depend on $T$ or $\mathcal{O}$, we expect it to approach a constant as $T \to \infty$ under mild distributional assumptions on $\|g_t\|_2$; see Appendix K. We finally note that although the results of this section are motivated by structured models, they apply to, and are novel for, the simpler models considered in previous work on ACV methods.

## 5 Experiments

We demonstrate the effectiveness of our proposed ACV methods on a diverse set of real-world examples where data exhibit temporal and spatial dependence: namely, temporal count modeling, named entity recognition, and spatial modeling of crime data. Additional experiments validating the accuracy and computational benefits afforded by LSCV are available in Appendix M.1, where we explore auto-regressive HMMs for motion capture analysis – with $N = 124$, $T$ up to 100, and $D$ up to 11,712.

**Approximate leave-within-sequence-out CV: Time-varying Poisson processes.** We begin by examining approximate LWCV (Algorithm 1) for maximum a posteriori (MAP) estimation. We consider a time-varying Poisson process model used by [Ihler et al., 2006] for detecting events in temporal

count data. We analyze loop sensor data collected every five minutes over a span of 25 weeks from a section of a freeway near a baseball stadium in Los Angeles. For this problem, there is one observed sequence ($N = 1$) with $T = 50{,}400$ total observations. There are $D = 11$ parameters. Full model details are in Appendix L.1.

To choose the folds in both exact CV and our ACV method, we consider two schemes, both following style (A) in Eq. (4); i.e., we omit observations (but not latents) in the folds. First, we follow the recommendation of Celeux and Durand [2008]; namely, we form each fold by selecting $m\%$ of measurements to omit (i.e., to form $\mathbf{o}$) uniformly at random and independently across folds. We call this scheme *i.i.d. LWCV*. Second, we consider a variant where we omit $m\%$ of observations in a contiguous block. We call this scheme *contiguous LWCV*; see Appendix L.1.

In evaluating the accuracy of our approximation, we focus on a subset of $T_{sub} = 10{,}000$ observations, plotted in the top panel of Fig. 1. The six panels in the lower left of Fig. 1 compare our ACV estimates to exact CV. Columns range over left-out percentages $m = 2, 5, 10$ (all on the data subset); rows depict i.i.d. LWCV (upper) and contiguous CV (lower). For each of $|\mathcal{O}| = 10$ folds and for each point $x_t$ left out in each fold, we plot a red dot with the exact fold loss $-\log p(x_t \mid \mathbf{x}_{[T]-\mathbf{o}}; \hat{\Theta}(\mathbf{w_o}))$ as its horizontal coordinate and our approximation $-\log p(x_t \mid \mathbf{x}_{[T]-\mathbf{o}}; \hat{\Theta}_{\mathrm{IJ}}(\mathbf{w_o}))$ as its vertical coordinate. We can see that every point lies close to the dashed black $x = y$ line; that is, the quality of our approximation is uniformly high across the thousands of points in each plot.

In the two lower right panels of Fig. 1, we compare the speed of exact CV to our approximation and the Newton step (NS) approximation [Beirami et al., 2017, Rad and Maleki, 2020] on two data subsets (size 5,000 and 10,000) and the full data. No reported times include the initial $\hat{\Theta}(\mathbf{1}_T)$ computation since $\hat{\Theta}(\mathbf{1}_T)$ represents the unavoidable cost of the data analysis itself. I.i.d. LWCV appears in the upper plot, and contiguous LWCV appears in the lower. For our approximation, we use 1,000 folds. Due to the prohibitive cost of both exact CV and NS, we run them for 10 folds and multiply by 100 to estimate runtime over 1,000 folds. We see that our approximation confers orders of magnitude in time savings both over exact CV and approximations based on NS. In Appendix I, we show that the approximations based on NS do not substantatively improve upon those provided by the significantly cheaper IJ approximations.

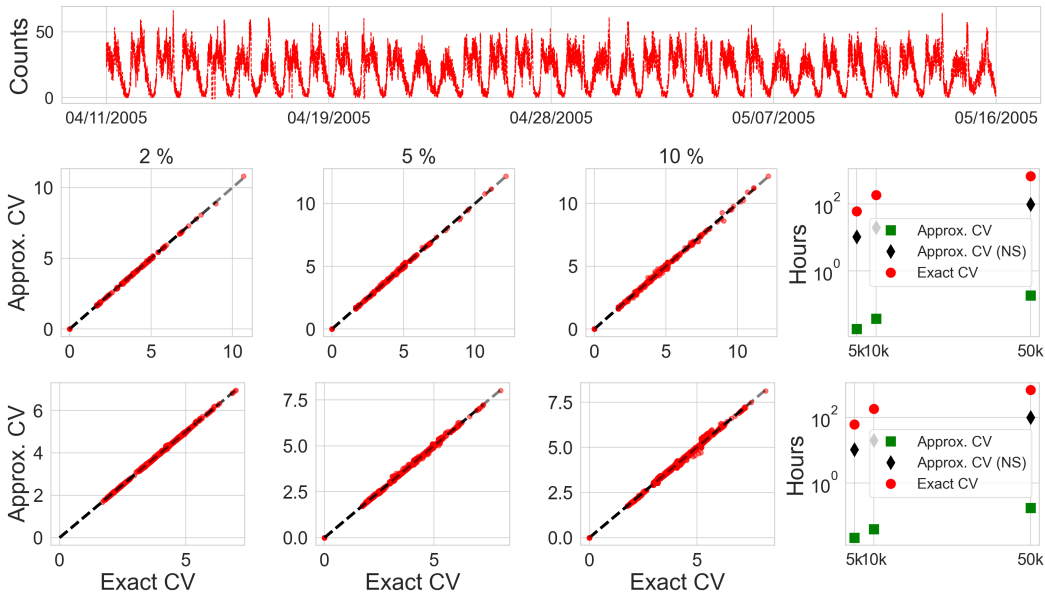

Figure 1: Evaluation of approximate LWCV for time-varying Poisson processes. *(Top panel)* A subset of the count series. *(Lower left six panels)* Scatter plots comparing exact CV loss (horizontal axis) at each point in each fold (red dots) to our approximation of CV loss (vertical axis). Black dashed line shows perfect agreement. Three columns for percent points left out; two rows for i.i.d. LWCV (upper) and contiguous LWCV (lower). *(Lower right two panels)* Wall-clock time for exact and approximate CV measured on a 2.5GHz quad core Intel i7 processor with 16GB of RAM; same rows as left panels.

**Robustness to inexact optimization: Neural conditional random fields.** Next, we examine the

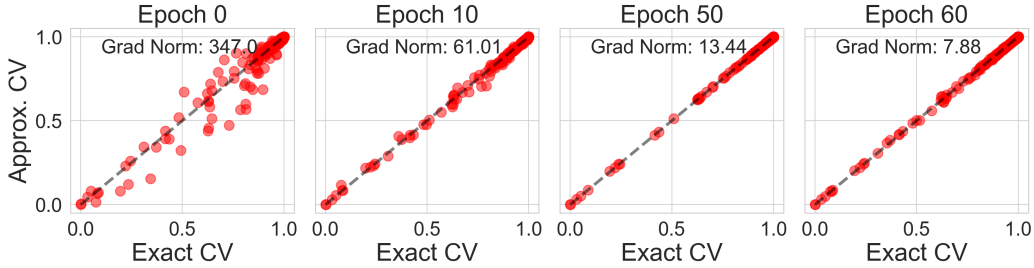

Figure 2: Behavior of ACV at different epochs in stochastic optimization for a bilstm-crf. Scatter plots comparing held out probabilities under CV (horizontal axis) at each point in each fold (red dots) to our approximation of CV (vertical axis). Black dashed line shows perfect agreement.

effect of using an inexact optimum $\Theta^{(S)}$, instead of the exact initial optimum $\hat{\Theta}(\mathbf{1}_T)$, as the input in our approximations. We consider LSCV for a bidirectional LSTM CRF (bilstm-crf) [Huang et al., 2015], which has been found [Lample et al., 2016, Ma and Hovy, 2016, Reimers and Gurevych, 2017] to perform well for named entity recognition. In this case, our problem is supervised; the words in input sentences ($\mathbf{x}_n$) are annotated with entity labels ($\mathbf{z}_n$), such as organizations or locations. We trained the bilstm-crf model on the CoNLL-2003 shared task benchmark [Sang and De Meulder, 2003] using the English subset of the data and the pre-defined train/validation/test splits containing 14,987(=$N$)/3,466/3,684 sentence annotation pairs. Here $T$ is the number of words in a sentence; it varies by sentence with a max of 113 and median of 9. The number of parameters $D$ is 99. Following standard practice, we optimize the full model using stochastic gradient methods and employ early stopping by monitoring loss on the validation set. See Appendix L.2 for model architecture and optimization details. In our experiments, we hold the other network layers (except for the CRF layer) fixed, and report epochs for training on the CRF layer after full-model training; this procedure mimics some transfer learning methods [Huh et al., 2016].

We consider 500 LSCV folds with one sentence (i.e., one $n$ index) per fold; the 500 points are chosen uniformly at random. The four panels in Fig. 2 show the behavior of our approximation (Algorithm 3 in Appendix G) at different training epochs during the optimization procedure. To ensure invertibility of the Hessian when far from an optimum, we add a small ($10^{-5}$) regularizer to the diagonal. At each epoch, for each fold, we plot a red dot with the exact fold held out probability $p(\mathbf{z}_n \mid \mathbf{x}_n; \hat{\Theta}(\mathbf{w}_{\{n\}})$ as its horizontal coordinate and our approximation $p(\mathbf{z}_n \mid \mathbf{x}_n; \hat{\Theta}_{\text{IJ}}(\mathbf{w}_{\{n\}})$ as the vertical coordinate. Note that the LSCV loss has no dependence on other $n$ due to the model independence across $n$; see Appendix G. Even in early epochs with larger gradient norms, every point lies close to the dashed black $x = y$ line. Fig. 5 of Appendix L.2 further shows the mean absolute approximation error between the exact CV held out probability and our approximation, across all 500 folds as a function of log gradient norm and wall clock time. As expected, our approximation has higher quality at better initial fits. Nonetheless, we see that decay in performance away from the exact optimum is gradual.

**Beyond chain-structured graphs: Crime statistics in Philadelphia.** The models in our experiments above are all chain-structured. Next we consider our approximations to LWCV in a spatial model with more complex dependencies. Balocchi and Jensen [2019], Balocchi et al. [2019] have recently studied spatial models of crime in the city of Philadelphia. We here consider a (simpler) hidden MRF model for exposition: a Poisson mixture with spatial dependencies, detailed in Appendix L.3. Here, there is a single structure observation ($N = 1$); there are $T = 384$ census tracts in the city; and there are $D = 2$ parameters. The data is shown in the upper lefthand panel of Fig. 3.

We choose one point per fold in style (B) of LWCV here, for a total of 384 folds. We test our method across four fixed values of a hyperparameter $\beta$ that encourages adjacent tracts to be in the same latent state. For each fold, we plot a red dot comparing the exact fold loss $-\log p(x_t \mid \mathbf{x}_{[T]-\{t\}}; \hat{\Theta}(\mathbf{w}_{\{t\}}))$ with our approximation $-\log p(x_t \mid \mathbf{x}_{[T]-\{t\}}; \hat{\Theta}_{\text{IJ}}(\mathbf{w}_{\{t\}}))$. The results are in the lower four panels of Fig. 3, where we see uniformly small error across folds in our approximation. In the upper right panel of Fig. 3, we see that our method is orders of magnitude faster than exact CV.

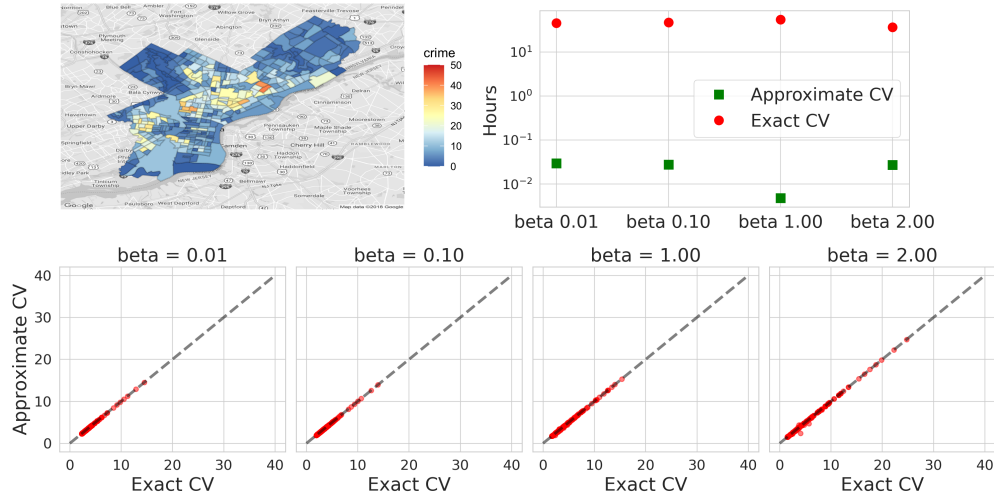

Figure 3: Evaluation of LWCV for loopy Markov random field. *(Top left)* Census tracts data. *(Upper right)* Wall-clock time of approximate CV and exact CV. *(Lower)* Scatter plots comparing CV loss (horizontal axis) at each point in each fold (red dots) to our approximation of CV loss (vertical axis). Black dashed line shows perfect agreement. Plots generated with different values of connectivity $\beta$.

**Discussion.** In this work, we have demonstrated how to extend approximate cross-validation (ACV) techniques to CV tasks with non-trivial dependencies between folds. We have also demonstrated that IJ approximations can retain their usefulness even when the initial data fit is inexact. While our motivation in the latter case was formed by complex models of dependent structures, our results are also applicable to, and novel for, the classic independence framework of ACV. An interesting remaining challenge for future work is to address other sources of computational expense in structured models. For instance, even after computing $\hat{\Theta}^{\backslash\mathbf{o}}$, inference can be expensive in very large graphical models; it remains to be seen if reliable and fast approximations can be found for this operation as well.

## Broader Impact

Accurate evaluation enables more reliable machine learning methods and more trustworthy communication of their capabilities. To the extent that machine learning methods may be beneficial – in that they may be used to facilitate medical diagnosis, assistive technology for individuals with motor impairments, or understanding of helpful economic interventions – accurate evaluation ensures these benefits are fully realized. To the extent that machine learning methods may be harmful – in that they may used to facilitate the spread of false information or privacy erosion – accurate evaluation should still make these methods more effective at their goals, even if societally undesirable. As in any machine learning methodology, it is also important for the buyer to beware; while we have tried to pick a broad array of experimental settings and to support our methods with theory, there may remain cases of interest when our approximations fail without warning. In fact, we take care to note that cross-validation and its points of failure are still not fully understood. All of our results are relative to exact cross-validation – since it is taken as the de facto standard for evaluation in the machine learning community (not without reason [Musgrave et al., 2020]). But when exact cross-validation fails, we therefore expect our method to fail as well.

**Acknowledgments.** This work was supported by the MIT-IBM Watson AI Lab, DARPA, the CSAIL–MSR Trustworthy AI Initiative, an NSF CAREER Award, an ARO YIP Award, ONR, and Amazon. Broderick Group is also supported by the Sloan Foundation, ARPA-E, Department of the Air Force, and MIT Lincoln Laboratory.

## Footnotes

[3]Note that the weight formulation could be extended to even more general reweightings in the spirit of the bootstrap. Exploring the bootstrap for structured models is outside the scope of the present paper.

[4]In practice, for numerical stability, we compute a Cholesky factorization of $H$.

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
