[Supplementary Material]

# A    Related work: Approximate CV methods

A growing body of recent work has focused on various methods for approximate CV (ACV). As outlined in the introduction, these methods generally take one of two forms. The first is based on taking a single Newton step on the leave-out objective starting from the full data fit, $\hat{\Theta}$. This approximation was first proposed by Obuchi and Kabashima [2016, 2018] for the special cases of linear and logistic regression and first applied to more general models by Beirami et al. [2017]. While this approximation is generally applicable to any CV scheme (e.g. beyond LOOCV) and any model type (e.g. to structured models), it is only efficiently applicable to LOOCV for GLMs. In particular, approximating each $\hat{\Theta}^{\backslash o}$ requires the computation and inversion of the leave-out objective's $D \times D$ Hessian matrix. In the case of LOOCV GLMs, this computation can be performed quickly using standard rank-one matrix updates; however, in more general settings, no such convenience applies.

Various works detail the theoretical properties of the NS approximation. Beirami et al. [2017], Rad and Maleki [2020] provide some of the first bounds on the quality of the NS approximation, but under fairly strict assumptions. Beirami et al. [2017] assume boundedness of both the parameter and data spaces, while Rad and Maleki [2020] require somewhat hard-to-check assumptions about the regularity of each leave-out objective (although they successfully verify their assumptions on a handful of problems). Koh et al. [2019] prove bounds on the accuracy of the NS approximation with fairly standard assumptions (e.g. Lipschitz continuity of higher-order derivatives of the objective function), but restricted to models using $\ell_2$ regularization. Wilson et al. [2020] also prove bounds on the accuracy of NS using slightly more complex assumptions but avoiding the assumption of $\ell_2$ regularization. More importantly, Wilson et al. [2020] also address the issue of model selection, whereas all previous works had focused on the accuracy of NS for model assessment (i.e. assessing the error of a single, fixed model). In particular, Wilson et al. [2020] give assumptions under which the NS approximation is accurate when used for hyperparameter tuning.

Finally, we note that in its simplest form, the NS approximation requires second differentiability of the model objective. Obuchi and Kabashima [2016, 2018], Rad and Maleki [2020], Beirami et al. [2017], Stephenson and Broderick [2020] propose workarounds specific to models using $\ell_1$-regularization. More generally, Wang et al. [2018] provide a natural extension of the NS approximation to models with either non-differentiable model losses or non-differentiable regularizers.

Again, while these NS methods can be applied to the structured models of interest here, the repeated computation and inversion of Hessian matrices brings their speed into question. To avoid this issue, we instead focus on approximations based on the infinitestimal jackknife (IJ) from the statistics literature [Jaeckel, 1972, Efron, 1981]. The IJ was recently conjectured as a potential approximation to CV by Koh and Liang [2017] and then briefly compared against the NS approximation for this purpose by Beirami et al. [2017]. The IJ was first studied in depth for approximating CV in an empirical and theoretical study by Giordano et al. [2019]. The benefit of the IJ in our application is that for any CV scheme[5] and any (differentiable and i.i.d.) model, the IJ requires only a single matrix inverse to approximate all CV folds. Koh et al. [2019] give further bounds on the accuracy of the IJ approximation for models using $\ell_2$ regularization. As in the case for NS, Wilson et al. [2020] give bounds on the accuracy of IJ beyond $\ell_2$ regularized models but with slightly more involved assumptions; Wilson et al. [2020] also give bounds on the accuracy of IJ for model selection.

Just as for the NS approximation, the IJ also requires second differentiability of the model objective. Stephenson and Broderick [2020] deal with this issue by noting that the methods of Wang et al. [2018] for applying the NS to non-differentiable objectives can be extended to cover the IJ as well. We note that the use of the IJ for model selection for non-differentiable objectives seems to be more complex than for the NS approximation. In particular, Stephenson and Broderick [2020, Appendix G] show that the IJ approximation can have unexpected and undesirable behavior when

used for tuning the regularization parameter for $\ell_1$ regularized models. Wilson et al. [2020] resolve this issue by proposing a further modification to the IJ approximation based on proximal operators.

## B Hidden Markov random fields

Here we show that HMMs are instances of (hidden) MRFs. Recall that a MRF models the joint distribution,

$$-\log p(\mathbf{x}, \mathbf{z}; \Theta) = -\sum_{n \in [N]} \log p(\mathbf{x}_n, \mathbf{z}_n; \Theta) = Z(\Theta) + \sum_{n=1}^{N} \left\{ \left[ \sum_{t \in [T]} \psi_t(x_{nt}, z_{nt}; \Theta) \right] + \left[ \sum_{\mathbf{c} \in \mathcal{F}} \phi_{\mathbf{c}}(z_{n\mathbf{c}}; \Theta) \right] \right\}. \tag{8}$$

**Hidden Markov models** We recover hidden Markov models as described in Section 2 by setting $\psi_t(x_{nt}, z_{nt}; \Theta) = \log F(x_{nt} \mid \theta_{z_{nt}})$, setting $\mathcal{F}$ to the set of all unary and pairwise indices, and defining $\phi_{t,t-1}(z_{nt}, z_{nt-1}; \Theta) = \log \text{Cat}(z_{nt} \mid A_{z_{nt-1}})$, and $\phi_1(z_{n1}) = \log \text{Cat}(z_{n1} \mid \pi)$, and $\phi_t(z_{nt}) = 0$, for $t \in [T] - 1$. The log normalization constant is $Z(\Theta) = 0$.

## C Leave structure out cross-validation (LSCV)

---
**Algorithm 2** Structured approximate LSCV
---
**Require:** $\Theta_1, \mathbf{x}, \mathcal{O}$
 1: Marginalize over $\mathbf{z}_n$: $\log p(\mathbf{x}_n; \Theta) = \text{MARG}(\mathbf{x}_n; \Theta), \forall n \in [N]$
 2: Compute $\log p(\mathbf{x}; \Theta, \mathbf{w}) = \sum_n w_n \log p(\mathbf{x}_n; \Theta)$
 3: Compute $H = \left. \frac{\partial^2 \log p(\mathbf{x}; \Theta, \mathbf{w}) + \log p(\Theta)}{\partial \Theta \partial \Theta^\top} \right|_{\Theta = \Theta_1, \mathbf{w} = \mathbf{1}_N}$
 4: Compute $J = (J_{dn}) := \left( \left. \frac{\partial^2 \log p(\mathbf{x}; \Theta, \mathbf{w}) + \log p(\Theta)}{\partial \Theta \partial w_n} \right|_{\Theta = \Theta_1, \mathbf{w} = \mathbf{1}_N} \right)$
 5: **for** $\mathbf{o} \in \mathcal{O}$, $\hat{\Theta}_{\text{ACV}}(\Theta_1, \mathbf{x}, \mathbf{o}) := \Theta_1 + \sum_{n \in \mathbf{o}} H^{-1} J_n$        # $J_n$ is $n$th column of $J$
 6: **return** $\{\hat{\Theta}_{\text{ACV}}(\Theta_1, \mathbf{x}, \mathbf{o})\}_{\mathbf{o} \in \mathcal{O}}$
---

## D Efficient weighted marginalization (WEIGHTEDMARG) for chain-structured MRFs

For chain-structured pairwise MRFs with discrete structure, we can use a dynamic program to efficiently marginalize out the structure. Assume that $z_t \; \forall t \in [T]$ can take one of $K$ values. Define $\alpha_{1k} = \exp[w_1 \psi_1(x_1, z_1 = k)]$, and then compute $\alpha_{tk}$ recursively:

$$\alpha_{t,k} = \sum_{\ell=1}^{K} \alpha_{t-1,\ell} \exp \left[ w_t \psi_t(x_t, z_t = k; \Theta) + \phi_{t,t-1}(z_t = k, z_{t-1} = \ell) \right], \tag{9}$$

if using weighting scheme (A) from Equation Eq. (4) or,

$$\alpha_{t,k} = \sum_{\ell=1}^{K} \alpha_{t-1,\ell} \exp \left[ w_t \psi_t(x_t, z_t = k; \Theta) + w_t w_{t-1} \phi_{t,t-1}(z_t = k, z_{t-1} = \ell) \right], \tag{10}$$

if using weighting scheme (B) from Equation Eq. (5). Then, for either (A) or (B), we have $p(\mathbf{x}; \Theta, \mathbf{w}) = \sum_{k=1}^{K} \alpha_{Tk}$. When $\mathbf{w} = \mathbf{1}_T$ we recover the empirical risk minimization solution. As is the case for non-weighted models, this recursion implies that $p(\mathbf{x}; \Theta, \mathbf{w})$ is computable in $O(TK^2Q)$ time instead of the usual $O(T^K Q)$ time required by brute-force summation (recall $Q$ is the time required to evaluate one local potential). Likewise, we can also compute the derivatives needed by Algorithm 1 in $O(TK^2Q)$ time either by manual implementation or automatic differentiation tools [Bartholomew-Biggs et al., 2000].

# E   Equivalence of weighting (A) and (B) for leave-future-out for chain-structured graphs

As noted in the main text, weighting schemes (A) and (B) are equivalent when the graph is chain structured. Formally,

**Proposition 3.** *Consider a chain-structured pairwise MRF with ordered indices $t$ on the chain (such as an HMM). Weighting styles (A) and (B) above are equivalent for* leave-future-out *CV. That is, choose $\mathbf{o} = \{T', \ldots, T\}$ for some $T' \in [T-1]$ (i.e., indices that are in the "future" when interpreted as time). Then set $\forall t \in \mathbf{o}, w_t = 0$ and $\forall t \in [T] - \mathbf{o}, w_t = 1$.*

This result does not hold generally beyond chain-structured graphical models – consider a four-node "ring" graph in which node $t$ is connected to nodes $t-1$ and $t+1$ (mod 4) for $t = 0, \ldots, 3$. Weighting scheme (B) produces a distribution that is chain-structured over three nodes, whereas (A) produces a distribution without such conditional independence properties. We now prove Proposition 3.

*Proof.* Recall that for a chain structured graph, we can write:

$$p(x, z) = p(x \mid z)p(z_1)\prod_{t=2}^{T} p(z_t \mid z_{t-1}).$$

Let $\mathbf{o} = \{T', T'+1, \ldots, T\}$ for some $T' < T$; that is, we are interested in dropping out time steps $T', \ldots, T$. For weighting scheme (A) (Eq. (4)), we drop out only the observations, obtaining:

$$p_A(x, z; w_\mathbf{o}) = \left(\prod_{t=1}^{T'-1} p(x_t \mid z_t)\right) p(z_1)\prod_{t=2}^{T} p(z_t \mid z_{t-1}).$$

When we sum out all $z$ to compute the marginal $p_A(x; w_\mathbf{o})$, we can first sum over $z_T, \ldots, z_{T'}$. As $\sum_{z_t} p(z_t \mid z_{t-1}) = 1$ for any value of $z_{T-1}$, we obtain:

$$p_A(x; w_\mathbf{o}) = \sum_{z_1, \ldots, z_{T'-1}} \left(\prod_{t=1}^{T'-1} p(x_t \mid z_t)\right) p(z_1) \prod_{t=2}^{T'-1} p(z_t \mid z_{t-1}),$$

which is exactly the formula for $p_B(x; w_\mathbf{o})$, the marginal likelihood from following weighting scheme (B) (Eq. (5)), in which we drop out both the $x_t$ and $z_t$ for $t \notin \mathbf{o}$.   □

# F   Conditional random fields

Conditional random fields assume that the *labels* $\mathbf{z}$ are observed and model the conditional distribution $p(\mathbf{z} \mid \mathbf{x}; \Theta)$. While more general dependencies between $\mathbf{x}$ and $\mathbf{z}$ are possible a commonly used variant [Ma and Hovy, 2016, Lample et al., 2016] captures the conditional distribution of the joint defined in Equation Eq. (8). Note,

$$
\begin{aligned}
\log p(\mathbf{z}_n \mid \mathbf{x}_n; \Theta) &= \log p(\mathbf{x}_n, \mathbf{z}_n; \Theta) - \log p(\mathbf{x}_n; \Theta) \\
&= -Z(\Theta) + \sum_{t \in [T]} \psi_t(x_{nt}, z_{nt}; \Theta) + \sum_{\mathbf{c} \in \mathcal{F}} \phi_\mathbf{c}(z_{n\mathbf{c}}; \Theta) \\
&\quad + Z(\Theta) - \int_{\mathbf{z}_n} \sum_{t \in [T]} \psi_t(x_{nt}, z_{nt}; \Theta) + \sum_{\mathbf{c} \in \mathcal{F}} \phi_\mathbf{c}(z_{n\mathbf{c}}; \Theta) d\mathbf{z}_n
\end{aligned}
\tag{11}
$$

Defining, $Z(\mathbf{x}_n; \Theta) = -\int_{\mathbf{z}_n} \sum_{t \in [T]} \psi_t(x_{nt}, z_{nt}; \Theta) + \sum_{\mathbf{c} \in \mathcal{F}} \phi_\mathbf{c}(z_{n\mathbf{c}}; \Theta) d\mathbf{z}_n$, then gives us the following conditional distribution,

$$
-\log p(\mathbf{z} \mid \mathbf{x}; \Theta) = \sum_{n=1}^{N} \left\{ Z(\mathbf{x}_n; \Theta) + \sum_{t \in [T]} \psi_t(x_{nt}, z_{nt}; \Theta) + \sum_{\mathbf{c} \in \mathcal{F}} \phi_\mathbf{c}(z_{n\mathbf{c}}; \Theta) \right\}.
\tag{12}
$$

Note that $Z(\mathbf{x}_n; \Theta)$ is an observation specific negative normalization constant.

## G  CV for conditional random fields

Analogously to the MRF case, we have two variants for CRFs — LSCV and LWCV. While LSCV is frequently used in practice, for example, [DeCaprio et al., 2007], we are unaware of instances of LWCV in the literature. Thus, while we derive approximations to both CV schemes, our CRF-based experiments in Section 5 only use LSCV.

### G.1  LSCV for CRFs

Leave structure out CV is analogous to the MRF case and is detailed in Algorithm 3, where $\log \tilde{p}(\mathbf{z}_n, \mathbf{x}_n; \Theta) := \sum_{t \in [T]} \psi_t(x_{nt}, z_{nt}; \Theta) + \sum_{\mathbf{c} \in \mathcal{F}} \phi_{\mathbf{c}}(z_{n\mathbf{c}}; \Theta)$. Since all input, label pairs $\{\mathbf{x}_n, \mathbf{z}_n\}$ are independent, $\log p(\mathbf{z} \mid \mathbf{x}; \Theta, \mathbf{w})$ is just a weighted sum across $n$ and the losses $-\log p(\mathbf{z}_n \mid \mathbf{x}_n; \hat{\Theta}(\mathbf{w}_{\{n\}}))$ and $-\log p(\mathbf{z}_n \mid \mathbf{x}_n; \hat{\Theta}_{IJ}(\mathbf{w}_{\{n\}}))$ do not depend on $[N] - n$.

---
**Algorithm 3** Structured approximate cross-validation (LSCV) for CRFs

**Require:** $\Theta_1, \mathbf{x}, \mathbf{z}, \mathcal{O}$
1: Compute $Z(\mathbf{x}_n; \Theta) = -\text{MARG}(\mathbf{x}_n; \Theta), \forall n \in [N]$
2: Compute $\log p(\mathbf{z} \mid \mathbf{x}; \Theta, \mathbf{w}) = \sum_n w_n \big[ Z(\mathbf{x}_n; \Theta) + \log \tilde{p}(\mathbf{z}_n, \mathbf{x}_n; \Theta) \big]$
3: Compute $H = \dfrac{\partial^2 \log p(\mathbf{z}|\mathbf{x}; \Theta, \mathbf{w}) + \log p(\Theta)}{\partial \Theta \partial \Theta^\top} \Big|_{\Theta = \Theta_1, \mathbf{w} = \mathbf{1}_N}$
4: Compute matrix $J := (J_{dn}) = \left( \dfrac{\partial^2 \log p(\mathbf{z}|\mathbf{x}; \Theta, \mathbf{w}) + \log p(\Theta)}{\partial \Theta \partial w_n} \Big|_{\Theta = \Theta_1, \mathbf{w} = \mathbf{1}_N} \right)$
5: **for** $\mathbf{o} \in \mathcal{O}$, $\hat{\Theta}_{\text{ACV}}(\Theta_1, \mathbf{x}, \mathbf{z}, \mathbf{o}) := \Theta_1 + \sum_{n \in \mathbf{o}} H^{-1} J_n$        # $J_n$ is $n$th column of $J$
6: **return** $\{\hat{\Theta}_{\text{ACV}}(\Theta_1, \mathbf{x}, \mathbf{z}, \mathbf{o})\}_{\mathbf{o} \in \mathcal{O}}$

---

### G.2  LWCV for CRFs

Leave within structure out for CRFs again comes with a choice of weighting scheme. Given a single input, label pair $\mathbf{x}, \mathbf{z}$, the $z_t$ are the outputs at location $t$, and the $x_t$ are the corresponding inputs. A form of CV arises when we drop the outputs $z_t$, for $t \in \mathbf{o}$. This gives us weighting scheme (C),

$$
\begin{aligned}
\hat{\Theta}(\mathbf{w}) = &\underset{\Theta}{\text{argmin}}\, Z(\Theta, \mathbf{w}, \mathbf{x}) \\
&+ \left[ \sum_{t \in [T]} w_t \psi_t(x_t, z_t; \Theta) + (1 - w_t) \int_{z_t} \psi_t(x_t, z_t; \Theta)\, dz_t \right] \\
&+ \left[ w_t \sum_{\mathbf{c} \in \mathcal{F}} \phi_{\mathbf{c}}(z_{\mathbf{c}}; \Theta) + (1 - w_t) \int_{z_t} \sum_{\mathbf{c} \in \mathcal{F}} \phi_{\mathbf{c}}(z_{\mathbf{c}}; \Theta)\, dz_t \right] - \log p(\Theta).
\end{aligned}
\tag{13}
$$

For linear chain structured CRFs with discrete outputs $\mathbf{z}$ a variant of the forward algorithm can be used to efficiently compute $Z(\Theta, \mathbf{w}, \mathbf{x})$ as well as the marginalizations over $\{z_t \mid t \in \mathbf{o}\}$ required by Eq. (13). See Bellare and McCallum [2007], Tsuboi et al. [2008] for details. Algorithm 4 summarizes the steps involved.

## H  Computational cost of one Newton-step-based ACV

Recall that we define $M$ to be the cost of one marginalization over the latent structure $z$ and noted above that the cost of computing the Hessian via automatic differentiation is $O(M)$. For the Newton step (NS) approximation, recall that we need to compute a different Hessian for each fold $\mathbf{o}$. While this can be avoided using rank-one update rules in the case of leave-one-out CV for generalized linear models, this is not the case for the CV schemes and models considered here. Thus, to use the Newton step approximation here, we require $O(M|\mathcal{O}|)$ time to compute all needed Hessians. Compared to the $O(M)$ time spent computing Hessians by our algorithms, the Newton step is significantly more expensive. For this reason, we do not consider Newton step based approximations here.

**Algorithm 4** Approximate leave-within-structure-out cross-validation for CRFs

**Require:** $\Theta_1, \mathbf{x}, \mathbf{z}, \mathcal{O}$
1: Compute *unweighted* marginalization over $\mathbf{z}$, $Z(\mathbf{x}; \Theta) = -\text{MARG}(\mathbf{x}_n; \Theta), \forall n \in [N]$
2: Compute *weighted* marginalization over $\mathbf{z}$: $Z(\mathbf{x}; \Theta, \mathbf{w}) = \text{WEIGHTEDMARG}(\mathbf{x}, \Theta, \mathbf{w})$.
3: Compute $\log p(\mathbf{z} \mid \mathbf{x}; \Theta) = Z(\mathbf{x}; \Theta, \mathbf{w}) + Z(\mathbf{x}_n; \Theta)$
4: Compute $H = \left. \frac{\partial^2 \log p(\mathbf{x}; \Theta, \mathbf{w}) + \log p(\Theta)}{\partial \Theta \partial \Theta^\top} \right|_{\Theta = \Theta_1, \mathbf{w} = \mathbf{1}_T}$
5: Compute matrix $J := (J_{dt}) = \left( \left. \frac{\partial^2 \log p(\mathbf{x}; \Theta, \mathbf{w}) + \log p(\Theta)}{\partial \Theta \partial w_t} \right|_{\Theta = \Theta_1, \mathbf{w} = \mathbf{1}_T} \right)$
6: **for** $\mathbf{o} \in \mathcal{O}$, **do:** $\hat{\Theta}_{\text{ACV}}(\Theta_1, \mathbf{x}, \mathbf{o}) := \Theta_1 + \sum_{t \in \mathbf{o}} H^{-1} J_t$      # $J_t$ is $t$th column of $J$
7: **return** $\{\hat{\Theta}_{\text{ACV}}(\Theta_1, \mathbf{x}, \mathbf{o})\}_{\mathbf{o} \in \mathcal{O}}$

## I   Comparison of approximations afforded by one Newton-step-based and IJ based ACV

We revisit the LWCV experiments in time varying Poisson processes described in Section 5. We agian focus on the $T_{sub} = 10,000$ subset of observations, plotted in the top panel of Fig. 1. In Fig. 4 we compare estimates provided by ACV based on one NS to those provided by IJ based ACV. The left plot depicts i.i.d LWCV and the right depicts contiguous LWCV when $m = 10\%$ of the subset is held out. Similar results hold for $m = 2\%$ and $m = 5\%$. For each of $|\mathcal{O}| = 10$ folds and for each point $x_t$ left out in each fold, we plot a red dot with the NS bases approximate fold loss as its horizontal coordinate and our IJ based approximation as its vertical coordinate. We can see that every point lies close to the dashed black $x = y$ line; that is, the quality of the two approximations largely agree across the thousands of points in each plot.

Figure 4: Comparison of NS and IJ approximate LWCV for time-varying Poisson processes. Scatter plots comparing NS based ACV loss (horizontal axis) at each point in each fold (red dots) to IJ based ACV loss (vertical axis). Black dashed line shows perfect agreement. Left plot containts i.i.d. LWCV results and the right plot contains contiguous LWCV results.

## J   Derivation of IJ approximations

In all cases considered here (i.e., the "exchangeable" leave-one-out CV considered by previous work or the more structured variants for chain-structured or general graph structured models) can be derived

similarly. In particular, once we have derived the relevant weighted optimization problem for each case, the derivation of the IJ approximation is the same. Let the relevant weighted optimization problem be defined for $w \in \mathbb{R}^T$:

$$\hat{\Theta}(w) := \underset{\Theta \in \mathbb{R}^D}{\operatorname{argmin}} \, F(\Theta, w),$$

where $F$ is some objective function with $F(\cdot, \mathbf{1}_T)$ corresponding to the "full-data" fit (i.e., without leaving out any data). We now follow the derivation of the IJ in Giordano et al. [2019]. The condition that $\hat{\Theta}(\mathbf{1}_T)$ is an exact optimum is:

$$\left. \frac{\partial F}{\partial \Theta} \right|_{\hat{\Theta}(\mathbf{1}_T), \mathbf{1}_T} = 0.$$

If we take a derivative with respect to $w_t$:

$$\left. \frac{\partial^2 F}{\partial \Theta \partial \Theta^T} \right|_{\hat{\Theta}(\mathbf{1}_T), \mathbf{1}_T} \left. \frac{d\Theta}{dw_t} \right|_{\hat{\Theta}(\mathbf{1}_T), \mathbf{1}_T} + \left. \frac{\partial^2 F}{\partial \Theta \partial w_t} \right|_{\hat{\Theta}(\mathbf{1}_T), \mathbf{1}_T} \left. \frac{dw_t}{dw_t} \right|_{\hat{\Theta}(\mathbf{1}_T), \mathbf{1}_T} = 0.$$

Noting that $dw_t / dw_t = 1$ and solving for $d\Theta / dw_t$:

$$\left. \frac{d\Theta}{dw_t} \right|_{\hat{\Theta}(\mathbf{1}_T), \mathbf{1}_N} = - \left( \left. \frac{\partial^2 F}{\partial \Theta \partial \Theta^T} \right|_{\hat{\Theta}(\mathbf{1}_T), \mathbf{1}_T} \right)^{-1} \frac{\partial^2 F}{\partial \Theta \partial w_t} \tag{14}$$

Thus we can form a first order Taylor series of $\hat{\Theta}(w)$ in $w$ around $w = \mathbf{1}_N$ to approximate:

$$\hat{\Theta}_{IJ}(w) \approx \hat{\Theta}(\mathbf{1}_T) - \sum_{t=1}^{T} \left( \left. \frac{\partial^2 F}{\partial \Theta \partial \Theta^T} \right|_{\hat{\Theta}(\mathbf{1}_T), \mathbf{1}_T} \right)^{-1} \frac{\partial^2 F}{\partial \Theta \partial w_t} (1 - w_t).$$

Specializing this last equation to the various $F$ and weight vectors $w$ of interest derives each of our ACV algorithms.

# K   Inexact optimization

We prove here a slightly more general version of Proposition 2 that covers both LWCV and LSCV, as well as arbitrary loss functions $\ell$. To encompass both in the same framework, let $\mathbf{w}_n \in \mathbb{R}^T$ be weight vectors for each structured object $n = 1, \ldots, N$. Our weighted objective will be:

$$\hat{\Theta}(\mathbf{w}) = \underset{\Theta \in \mathbb{R}^D}{\operatorname{argmin}} \sum_{n=1}^{N} \log p(\mathcal{D}_n; \Theta, \mathbf{w}_n) + p(\Theta),$$

where $\mathcal{D} = \{\mathcal{D}_1, \ldots, \mathcal{D}_N\}$ denotes the collection of all observed structures; i.e., each $\mathcal{D}_n$ may be a sequence of observations $x_n$ for a HMM or observed outputs and inputs $x_n, z_n$ for a CRF. Let $\hat{\Theta}(\mathbf{1}_{NT})$ be the solution to this problem with $w_{nt} = 1$ for all $n$ and $t$. We assume that we are interested in estimating the exact out-of-sample loss for some generic loss $\ell$ by using exact CV, $\mathcal{L}_{\mathrm{CV}} := (1/|\mathcal{O}|) \sum_{\mathbf{o}} \ell(\mathcal{D}_{\mathbf{o}}, \mathcal{D}_{-\mathbf{o}}, \hat{\Theta}(\mathbf{w}_{\mathbf{o}}))$; e.g., we may have $\ell(\mathcal{D}_{\mathbf{o}}, \mathcal{D}_{-\mathbf{o}}, \hat{\Theta}(\mathbf{w}_{\mathbf{o}})) = -\log p(x_{\mathbf{o}} \mid x_{[T]-\mathbf{o}}; \hat{\Theta}(\mathbf{w}_{\mathbf{o}}))$ in the case of a HMM with $N = 1$. Notice here that $\mathbf{o} \subset [N] \times [T]$ indexes arbitrarily across structures. We can now state a modified version of Assumption 4.

**Assumption 5.** *Let $B \subset \mathbb{R}^D$ be a ball centered on $\hat{\Theta}(\mathbf{1}_{NT})$ and containing $\Theta^{(S)}$. Then the objective $\sum_n \log p(x_n; \Theta, \mathbf{1}_T) + p(\Theta)$ is strongly convex with parameter $\lambda_{\min}$ on $B$. Additionally, on $B$, the derivatives $g_{nt}(\Theta) := \partial^2 \log p(x_n; \Theta, \mathbf{w}_n)/\partial \Theta \partial w_{nt}$ are Lipschitz continuous with constant $L_g$ for all $n, t$ and the inverse Hessian of the objective is Lipschitz with parameter $L_{Hinv}$. Finally, on $B$, $\ell(\mathcal{D}_{\mathbf{o}}, \mathcal{D}_{-\mathbf{o}}, \Theta)$ is a Lipschitz function of $\Theta$ with parameter $L_\ell$ for all $\mathbf{o}$.*

We now prove our more general version Proposition 2.

**Proposition 4.** *Take Assumption 5. Then the approximation error of $\mathcal{L}_{\mathrm{IJ}}(\Theta^{(S)})$ is bounded by:*

$$|\mathcal{L}_{\mathrm{IJ}}(\Theta^{(S)}) - \mathcal{L}_{\mathrm{CV}}| \leq C\varepsilon_\Theta + \varepsilon_{\mathrm{IJ}}, \tag{15}$$

*where $C$ is given by*

$$\left( L_\ell + \frac{L_\ell L_g}{\lambda_{\min}} + \frac{L_\ell L_{Hinv}}{|\mathcal{O}|} \sum_{\mathbf{o}} \left\| \sum_{t \in \mathbf{o}} g_{nt}(\hat{\Theta}(\mathbf{1}_{NT})) \right\|_2 \right).$$

*Proof.* By the triangle inequality:
$$|\mathcal{L}_{\mathrm{IJ}}(\Theta^{(S)}) - \mathcal{L}_{\mathrm{CV}}| \leq$$
$$|\mathcal{L}_{\mathrm{IJ}}(\Theta^{(S)}) - \mathcal{L}_{\mathrm{IJ}}(\hat{\Theta}(\mathbf{1}_{NT}))|$$
$$+ |\mathcal{L}_{\mathrm{IJ}}(\hat{\Theta}(\mathbf{1}_{NT})) - \mathcal{L}_{\mathrm{CV}}|.$$

The second term is just the constant $\varepsilon_{\mathrm{IJ}}$. Now we just need to bound the first term using our Lipschitz assumptions. We have, by the triangle inequality

$$|\mathcal{L}_{\mathrm{IJ}}(\hat{\Theta}(\mathbf{1}_{NT})) - \mathcal{L}_{\mathrm{IJ}}(\Theta^{(S)})|$$
$$\leq \frac{1}{|\mathcal{O}|} \sum_{\mathbf{o}} \left| \ell\left( \mathcal{D}_{\mathbf{o}}, \mathcal{D}_{-\mathbf{o}}, \hat{\Theta}(\mathbf{1}_{NT}) + H^{-1}(\hat{\Theta}(\mathbf{1}_{NT})) \sum_{t \in \mathbf{o}} g_{nt}(\hat{\Theta}(\mathbf{1}_{NT})) \right) \right.$$
$$\left. - \ell\left( \mathcal{D}_{\mathbf{o}}, \mathcal{D}_{-\mathbf{o}}, \Theta^{(S)} + H^{-1}(\Theta^{(S)}) \sum_{t \in \mathbf{o}} g_{nt}(\Theta^{(S)}) \right) \right|.$$

Continuing to apply the triangle inequality and our Lipschitz assumptions:

$$\leq \frac{L_\ell}{|\mathcal{O}|} \sum_{\mathbf{o}} \left( \left\| \hat{\Theta}(\mathbf{1}_{NT}) - \Theta^{(S)} \right\|_2 + \left\| H^{-1}(\hat{\Theta}(\mathbf{1}_{NT})) \sum_{t \in \mathbf{o}} g_{nt}(\hat{\Theta}(\mathbf{1}_{NT})) - H^{-1}(\Theta^{(S)}) \sum_{t \in \mathbf{o}} g_{nt}(\Theta^{(S)}) \right\|_2 \right)$$
$$\leq L_\ell \varepsilon_\Theta + \frac{L_\ell}{|\mathcal{O}|} \sum_{\mathbf{o}} \left\| H^{-1}(\Theta^{(S)}) \sum_{t \in \mathbf{o}} \left( g_{nt}(\hat{\Theta}(\mathbf{1}_{NT})) - g_{nt}(\Theta^{(S)}) \right) \right\|_2$$
$$+ \frac{L_\ell}{|\mathcal{O}|} \sum_{\mathbf{o}} \left\| \left( H^{-1}(\hat{\Theta}(\mathbf{1}_{NT})) - H^{-1}(\Theta^{(S)}) \right) \sum_{t \in \mathbf{o}} g_{nt}(\hat{\Theta}(\mathbf{1}_{NT})) \right\|_2$$
$$\leq \left( L_\ell + \frac{L_\ell L_g}{\lambda_{\min}} + \frac{L_\ell L_{Hinv}}{|\mathcal{O}|} \sum_{\mathbf{o}} \left\| \sum_{t \in \mathbf{o}} g_{nt}(\hat{\Theta}(\mathbf{1}_{NT})) \right\|_2 \right) \varepsilon_\Theta.$$

Defining the term in the parenthesis as $C$ finishes the proof.

$\square$

As noted after the statement of Proposition 2 in the main text, $(1/|\mathcal{O}|) \sum_{\mathbf{o} \in \mathcal{O}} \left\| \sum_{t \in \mathbf{o}} g_{nt}(\hat{\Theta}(\mathbf{1}_{NT})) \right\|_2$ may depend on $T$, $N$ or $\mathcal{O}$, but we expect it to converge to a constant given reasonable distributional assumptions on the data. To build intuition, we consider the case of leave-one-out CV for generalized linear models, where we observe a dataset of size $N > 1$ and have $T = 1$. In particular, we have $\log(x_n, y_n; \Theta) = f(x_n^T \Theta, y_n)$, where $x_n \in \mathbb{R}^D$ are the covariates and $y_n \in \mathbb{R}$ are the responses. In this case, $g_{nt} = D_n^{(1)} x_n$, where $D_n^{(1)} = df(z)/dz \big|_{z = x_n^T \hat{\Theta}(\mathbf{1}_T)}$. Then, given reasonable distributional assumptions on the covariates and some sort of control over the derivatives $D_n^{(1)}$, we might suspect that $(1/N) \sum_n |D_n^{(1)}| \|x_n\|_2$ will converge to a constant. As an example, we consider logistic regression with sub-Gaussian data, for which we can actually prove high-probability bounds on this sum.

**Definition 1.** *[e.g., Vershynin [2018]] For $c_x > 0$, a random variable $V$ is $c_x$-sub-Gaussian if*
$$\mathbb{E}\left[ \exp\left( V^2/c_x^2 \right) \right] \leq 2.$$

**Proposition 5.** *For logistic regression, assume that the components of the covariates $x_{nd}$ are i.i.d. from a zero-mean $c_x$-sub-Gaussian distribution for $d = 1, \ldots, D$. Then we have that, for any $t \geq 0$:*

$$\Pr\left[ |\frac{1}{N} \sum_{n=1}^{N} \left\| \nabla f(\hat{\Theta}(\mathbf{1}_T), x_n) \right\|_2 - \sqrt{D}| \geq t \right] \leq \exp\left[ -C \frac{Nt^2}{c_x^2} \right], \tag{16}$$

*where $C > 0$ is some global constant, independent of $N$, $D$, and $c_x$.*

*Proof.* First, we can use the fact that $\left\| \nabla f(\hat{\Theta}(\mathbf{1}_T), x_n) \right\|_2 \leq \|x_n\|_2$, as for logistic regression, $|D_n^{(1)}| \leq 1$. Next, we can use the fact that $\|x_n\|_2 - \sqrt{D}$ is a zero-mean sub-Gaussian random variable by Theorem 3.1.1 of Vershynin [2018]. We can then apply Hoeffding's inequality [Vershynin, 2018, Theorem 2.6.3] to complete the proof. $\square$

# L   Experimental details

We provide further experimental details in this section.

## L.1   Time varying Poisson processes

We briefly summarize the time-varying Poisson process model from Ihler et al. [2006] here. Our data is a time series of loop sensor data collected every five minutes over a span of 25 weeks from a section of a freeway near a baseball stadium in Los Angeles. In all, there are 50,400 measurements of the number of cars on that span of the freeway. Ihler et al. analyze the resulting time series of counts $\mathbf{x}$ to detect the presence or absence of an event at the stadium. Following their model, we use a background Poisson process with a time varying rate parameter $\lambda_t$ to model *non-event* counts, $x_{b_t} \sim \text{Poisson}(\lambda_t)$. To model the daily variation apparent in the data, we define $\lambda_t \triangleq \lambda_o \delta_{d_t}$, where $d_t$ takes one of seven values, each corresponding to one day of the week and $[\delta_1/7, \dots, \delta_7/7] \sim \text{Dir}(1, \dots, 1)$. We use binary latent variables $z_t$ indicate the presence or absence of an event and assume a first order Markovian dependence, $z_t \mid z_{t-1} \sim A_{z_{t-1}}$. Next, $z_t = 0$ indicates a non-event at time step $t$ and the observed counts are generated as $x_t = x_{b_t}$. An event at time step $t$ corresponds to $z_t = 1$ and $x_t = x_{b_t} + x_{e_t}$, and $x_{e_t} \sim \text{NegBinomial}(x_{e_t} \mid a, b/(1+b))$, where $x_{e_t}$ are unobserved excess counts resulting from the event. We place Gamma priors on $\lambda_0, a, b$ and Beta priors on $A_{00}$ and $A_{11}$, and learn the MAP estimates of the parameters $\Theta = \{\lambda_0, \delta_1, \dots, \delta_7, a, b, A\}$ while marginalizing $x_{e_t}$ and $z_1, \dots, z_T$. We refer the interested reader to Ihler et al. [2006] for further details about the model and data.

**Contiguous LWCV.**   In contiguous LWCV we leave out contiguous blocks from a time series. To drop $m\%$ of the data, we sample an index $t$ uniformly at random from $[\lfloor mT/100 \rfloor + 1, \dots, T]$ and set $\mathbf{o} = \{t - \lfloor mT/100 \rfloor, \dots t\}$.

**Numerical values from Fig. 1**   In Table 1 we present an evaluation of the LWCV approximation quality for time-varying Poisson processes. The results presented are a numerical summary of the results visually illustrated in Fig. 1. Table 2 presents the wall clock time numbers plotted in the lower

|            | 2 %             | 5%              | 10 %            |
|------------|-----------------|-----------------|-----------------|
| i.i.d      | $0.005 \pm 0.009$ | $0.006 \pm 0.01$ | $0.006 \pm 0.005$ |
| contiguous | $0.003 \pm 0.003$ | $0.007 \pm 0.02$ | $0.007 \pm 0.006$ |

Table 1: Evaluation of approximate LWCV for time-varying Poisson processes. Mean ACV relative error, $|acv - cv|/cv$ and two standard deviations, over ten folds with $T = 10000$. The numbers summarize the scatter plots in the lower left six panels of Fig. 1. The column headers indicate the percentage of data in the held out fold.

right panels of Fig. 1.

|       | i.i.d |         |         | contiguous |         |         |
|-------|-------|---------|---------|------------|---------|---------|
| T     | ACV   | ACV (NS)  | Exact CV  | ACV     | ACV (NS) | Exact CV  |
| 5000  | 1.1 mins | 10.5 hours | 61.1 hours | 1.3 mins | 10.5 hours | 61.3 hours |
| 10000 | 2.2 mins | 19.9 hours | 185.8 hours | 2.4 mins | 19.9 hours | 182.4 hours |
| 50000 | 11.0 mins | 98.6 hours | 682.2 hours | 10.6 mins | 99.1 hours | 683.9 hours |

Table 2: Wall clock time from the two lower right panels in Fig. 1 at $T = 50000$ and with $m\% = 10\%$ of the data in the held out fold.

## L.2   Neural CRF

We employed a bi-directional LSTM model with a CRF output layer. We used a concatenation of a 300 dimensional Glove word embeddings [Pennington et al., 2014] and a character CNN [Ma and Hovy, 2016] based character representation. We employed variational dropout with a dropout rate of $0.25$. The architecture is detailed below.

Figure 5: *(Left panel)* Error in our approximation relative to exact CV averaged across folds, as a function of wall clock time. *(Right panel)* Error in our approximation relative to exact CV and averaged across folds, as a function of log gradient norm in the optimization procedure.

```
LSTMCRFVD(
  (dropout): Dropout(p=0.25, inplace=False)
  (char_feats_layer): CharCNN(
    (char_embedding): CharEmbedding(
      (embedding): Embedding(96, 50, padding_idx=0)
      (embedding_dropout): Dropout(p=0.25, inplace=False)
    )
    (cnn): Conv1d(50, 30, kernel_size=(3,), stride=(1,), padding=(2,))
  )
  (word_embedding): Embedding(2196016, 300)
  (rnn): StackedBidirectionalLstm(
    (forward_layer_0): AugmentedLstm(
      (input_linearity): Linear(in_features=330, out_features=200, bias=False)
      (state_linearity): Linear(in_features=50, out_features=200, bias=True)
    )
    (backward_layer_0): AugmentedLstm(
      (input_linearity): Linear(in_features=330, out_features=200, bias=False)
      (state_linearity): Linear(in_features=50, out_features=200, bias=True)
    )
    (forward_layer_1): AugmentedLstm(
      (input_linearity): Linear(in_features=100, out_features=200, bias=False)
      (state_linearity): Linear(in_features=50, out_features=200, bias=True)
    )
    (backward_layer_1): AugmentedLstm(
      (input_linearity): Linear(in_features=100, out_features=200, bias=False)
      (state_linearity): Linear(in_features=50, out_features=200, bias=True)
    )
    (layer_dropout): InputVariationalDropout(p=0.25, inplace=False)
  )
  (rnn_to_crf): Linear(in_features=100, out_features=9, bias=True)
  (crf): ConditionalRandomField()
)
```

**Training** We used Adam for optimization. Following the recommendation of Reimers and Gurevych [2017] we used mini-batches of size 31 Reimers and Gurevych [2017]. We employed early stopping by monitoring the loss on the validation set. Freezing all but the CRF layers we further fine-tuned only the CRF layer for an additional 60 epochs. In Fig. 5 we plot the mean absolute approximation error in the held out probability under exact CV and our approximation across all 500 folds as a function of (wall clock) time taken by the optimization procedure.

### L.3 Philadelphia crime experiment

Our crime data comes from opendataphilly.org, where the Philadelphia Police Department publicly releases the time, type, and location of every reported time. For each census tract, we have a latent label $z_t \in \{-1, 1\}$, and model the number of reported crimes $x_t$ with a simple Poisson mixture model: $x_t | z_t \sim \text{Poisson}(\lambda_{z_t})$ where $\lambda_{-1}, \lambda_1 > 0$ are the unknown mean levels of crime in low- and high-crime areas, respectively. Since we might expect adjacent census tracts to be in the same latent state, we model the $z_t$'s with an MRF so that

$$\log p(\mathbf{x}, \mathbf{z}; \Theta) = \sum_t \left[ -\lambda_{z_t} + x_t \log \lambda_{z_t} - \log(x_t!) \right] + \beta \sum_t \sum_{t' \in \Gamma(t)} \mathbf{1}\{z_t = z_{t'}\} - \log Z(\beta)$$

where $\Theta = \{\lambda_{-1}, \lambda_1\}$, $\Gamma(t)$ is the collection of census tracts that are spatially adjacent to census tract $t$ and $\log Z(\beta)$ is the log normalizer for the latent field $p(\mathbf{z})$. The potential $\mathbf{1}\{z_t = z_{t'}\}$ expresses prior belief that adjacent census tracts should be in the same latent class. The connection strength $\beta$ is treated as a hyper-parameter. For each $\beta$ fixed, $\Theta$ is estimated using expectation maximization Dempster et al. [1977] on $\sum_{\mathbf{z}} \log p(\mathbf{x}, \mathbf{z}; \Theta)$. M-step computation is analytical, given the posteriors $p(\mathbf{z}_t | \mathbf{x}; \Theta)$. Exact E-step computation is reasonably efficient through smart variable elimination [Koller and Friedman, 2009, Chapter 9]: the number of states is small and common heuristics to find good elimination orderings, such as MinFill, worked well. This efficient variable elimination order is also used to implement the WEIGHTEDMARG routine of 1.

## M    Additional experiments

We present additional experimental validation in support of the ACV methods in this section.

### M.1    Motion capture analysis

**Data.** We analyze motion capture recordings from the CMU MoCap database (http://mocap.cs.cmu.edu), which consists of several recordings of subjects performing a shared set of activities. We focus on the 124 sequences from the "Physical activities and Sports" category that has been previously been studied [Fox et al., 2009, Hughes et al., 2012, Fox et al., 2014] in the context of unsupervised discovery of shared activities from the observed sequences. At each time step we retain twelve measurements deemed informative for describing the activities of interest, as recommended by Fox et al. Auto-regressive hidden Markov models have been shown effective for this task, motivating their use in this section.

**Accurate LSCV— auto-regressive HMMs**    We confirm here that ACV is accurate and computationally efficient for structured models in the case studied by previous work: LSCV *with* exact model fits. We present comparisons between embarrassingly parallel exact CV and LSCV with parallelized Hessian computation ("Approx. Parallel", i.e., we parallelize the Hessian computation over different structures $n$), alleviating the primary computational bottleneck for ACV. We model the collection of MoCAP sequences via a $K$-state HMM with an order-p auto-regressive (AR(p)) observation model. We also consider variants where each state's auto-regressive model is parameterized via a neural network. Figure 6 visualizes a MoCAP sequence where we have retained only the 12 relevant dimensions. For this experiment, we retain up to $100(= T)$ measurements per sequence. We employ the following auto-regressive observation model,

$$p(x_{nt} \mid x_{nt-1}, \ldots, x_{nt-p}, z_{nt}) = \mathcal{N}(x_{nt} \mid \sum_{m=1}^{p} B_{z_{nt}} x_{nt-m} + b_{z_{nt}}, \sigma^2 \mathbf{I}),$$

$$B_k \sim \text{Matrix-Norm}(\mathbf{I}, \mathbf{I}, \mathbf{I}), \quad b_k \sim \mathcal{N}(0, \mathbf{I}) \quad \forall k \in \{1, \ldots, K\}, \tag{17}$$

where $p$ is the order of the auto-regression. Neural auto-regressive observation models are defined as,

$$p(x_{nt} \mid x_{nt-1}, \ldots, x_{nt-p}, z_{nt}) = \mathcal{N}(x_{nt} \mid B_{z_{nt}}^1 h(\sum_{m=1}^{p} B_{z_{nt}}^0 x_{nt-m} + b_{z_{nt}}^0) + b_{z_{nt}}^1, \sigma^2 \mathbf{I}),$$

$$\theta_k \sim \mathcal{N}(0, \lambda \mathbf{I}), \quad \forall k \in \{1, \ldots, K\}, \tag{18}$$

where $\theta_k = \{B_k^0, b_k^0, B_k^1, b_k^1\}$, and $h$ denotes a tanh non-linearity, and $B_k^0, B_k^0 \in \mathbb{R}^{12 \times 12}$ and $b_k^0, b_k^1 \in \mathbb{R}$, i.e., a 12-12-12 fully connected network.

Figure 6: Motion capture analysis through auto-regressive HMMs. *(Top)* A twelve dimensional MoCap sequence that serves as the observed data and the number of parameters $D$ for different models under consideration. The high dimensionality of the models make alternate ACV methods based on a single Newton step infeasible.*(Middle)* Scatter plots comparing leave one out loss, where x-axis is $-\ln p(\mathbf{x}_n \mid \Theta(\mathbf{w}_{\{n\}}))$ and y-axis is $-\ln p(\mathbf{x}_n \mid \hat{\Theta}_{\mathrm{IJ}}(\mathbf{w}_{\{n\}}))$ for different auto-regressive orders under exact and IJ approximated leave one out cross validation. Points along the diagonal indicate accurate IJ approximations. *(Bottom)* Timing and held out negative log probability across different models. For IJ and Exact the error bars represent two jackknife standard error. The IJ approximations are significantly faster but closely approximate exact leave one out loss across models and track well with test loss computed on the held out $20\%$ of the dataset.

While past work has explored AR(0) and AR(1) observation models, a thorough exploration of the effect of $p$ has been lacking. ACV provides an effective tool for exploring such questions accurately and inexpensively. We split the sequences into a $80/20\%$ train and test split and perform LSCV on the training data ($N = 100$) to compare AR(p) models with $p$ ranging from zero through five and the neural variant with $p = 1$ (NAR(1)), in terms of how well they describe the left out sequence. Following Fox et al., we fix $K = 16$. Figure 6 summarizes our results. First, we see that the ACV loss is quite close to the exact CV loss and that both track well with the held-out test loss. Furthermore, consistent with previous studies, we find that using an AR(1) observation model is significantly better than using an AR(0) or higher-order AR model. Interestingly, the out-of-sample loss for the AR(1) model is comparable to neural variant, NAR(1).

In terms of computation, the ACV is significantly faster than exact CV. In fact, for the higher order auto-regressive likelihoods and the neural variant, exact CV was too expensive to perform. Instead, we report estimated time for running such experiments by multiplying the average time taken to run three folds of LSCV with the number of training instances. For AR(0) and AR(1) we compare against exact CV implemented via publicly available optimized Expectation Maximization code [Hughes and Sudderth, 2014]. The higher order AR and the NAR(1) model, were fit by BFGS as implemented in SCIPY.OPTIMIZE.MINIMIZE. We find that computing the embarrassingly parallel version provides significant speedups over their serial counterparts.

Figure 7: Within sequence leave out experiments. We took the longest MoCAP sequence containing 1484 measurements and fit a five state HMM with Gaussian emissions. We find that even for the MoCAP data IJ approximations to i.i.d. LWCV is very accurate. As the contiguous LWCV involves making larger scale changes to the sequence, for instance at $10\%$ we end up dropping chunks of 140 time steps from the sequences, resulting in larger changes to the parameters, IJ approximations are relatively less accurate. *(Top)* Scatter plots comparing i.i.d LWCV loss $-\ln p(x_t \mid \mathbf{x}_{[T]-\mathbf{o}}; \Theta(\mathbf{w_o}))$ (horizontal axis) with $-\ln p(x_t \mid \mathbf{x}_{[T]-\mathbf{o}}; \hat{\Theta}_{\text{IJ}}(\mathbf{w_o}))$ (vertical axis), for each point $x_t$ left out in each fold, computed under exact CV for different omission rates $m\% = 2\%, 5\%$, and $10\%$ on $M = 10$ trials. *(Bottom)* Results for contiguous LWCV.

**Accurate LWCV for MoCAP**    Next, we present LWCV results on a $1{,}484$ measurement long sequence extracted from the MoCAP dataset. We explore three variants of LWCV: i.i.d LWCV, contiguous LWCV, and a special case of contiguous LWCV: leave-future-out CV. Figures 7 and 8 present these results. We find that the IJ approximations again provide accurate approximations to exact CV. The performance deteriorates for contiguous LWCV when large chunks of the sequence are left out. Since large changes to the sequence result in large changes to the fit parameters, a Taylor series approximation about the original fit is less accurate. Also, for high dimensional models such as NAR(1) IJ approximations tend to be less accurate [Stephenson and Broderick, 2020], explaining the drop in LFOCV performance for the NAR(1) model.

Figure 8: Leave Future Out CV for MoCAP data on a single MoCAP sequence containing 1484 measurements. The scatter plots compare $-\ln p(x_{T'} \mid \mathbf{x}_{[T]-\mathbf{o}}; \Theta(\mathbf{w_o}))$ (horizontal axis)) with $-\ln p(x_{T'} \mid \mathbf{x}_{[T]-\mathbf{o}}; \hat{\Theta}_{IJ}(\mathbf{w_o}))$ (vertical axis), with $\mathbf{o} = \{T', T'+1, \ldots T\}$, for some $T' \leq T$, for a five state HMM with Gaussian emissions (left), order 1 auto-regressive emissions (middle), neural auto-regressive emissions (right). The rightmost plot shows the number of parameters in each model. We vary $T'$ from $1337$ to $1484$ for Gaussian and AR(1) emissions. Since exact fits the NAR model are more expensive we only vary $T'$ between $1455$ and $1484$ for NAR(1). We find that ACV to be accurate. The NAR model which is an instance of a higher dimensional optimization problem, leads to approximations that are less accurate than the lower dimensional AR(0) and AR(1) cases.

## Footnotes

[5]The methods of Giordano et al. [2019] apply beyond CV to other "reweight and retrain" schemes such as the bootstrap. The methods presented in our paper apply more generally as well, although we do not explore this extension.