[Reviews · NeurIPS 2020]

Review 1

Summary and Contributions: The paper discusses fast approximation to cross-validation for Markov and conditional random field models when marginal maximum likelihood estimation is used for inference.

Strengths: For the type of models and inference, the proposed approach is sensible extension of the previous work. The experiments demonstrate the practically useful gain computation time, which will make it more feasible to use more elaborate models for bigger data sets.

Weaknesses: The limitations when the approach is applicable are not well stated. Marginal maximum likelihood is mentioned in the end of section 2. At this point I assumed that exact gradients and deterministic optimization would be used. In one of the experiments stochastic gradient methods is used, but it's not clearly stated whether stochastic gradients can be used also in Algorithm 1. This point affects my overall conclusion a lot. I expect that the authors may have good answer clarifying the conditions when the approach is applicable which they could include in the final version.

Correctness: Mostly correct with some misleading wording used as explained below. "But this existing ACV work is restricted to simpler models by the assumptions that (i) data are independent and (ii) an exact initial model fit is available. In structured data analyses, (i) is always untrue, and (ii) is often untrue." This is slightly misleading. If we assume complete independence, there is no common model. It would be better to discuss conditional independence. In case of structured models there can also be conditional independence (see, e.g. Figure 1 in https://arxiv.org/abs/1810.10559). Having a structured model, does not automatically invalidate, for example, leave-one-out cross-validation. The paper is now missing the distinction between the model structure and the prediction task, such that given certain prediction task, something like leave-one-out cross-validation may be inappropriate. For example, in case of time series data and structured model for the time dependency, the observations can be conditionally independent given the latent values, and leave-one-out cross-validation is valid, but it is the prediction task reasons why to leave out all observations at the end of a time series (see, e.g. discussion in B├╝rkner et al, 2020 cited in your paper and https://doi.org/10.1007/s42113-018-0020-6). Thus, it would be more accurate to write "approximate cross-validation for structured prediction in structured models". I don't mean you should change the title, but I suggest to consider some clarifications of this issue especially in the abstract and the introduction and some sentences later. The first paragraph of section 3 in your paper has discussion about extrapolation of time series and pictures and this discussion is towards the correct direction in sense of acknowledging that the structure in the prediction task is the crucial point why leave-one-out is not sufficient. You could use that example already in the introduction to introduce the concept of the prediction ta

Clarity: Paper is mostly well written. The title, the abstract and the introduction all mention that the approach is for "structured models", but how to approximate CV depends also on how the inference is made and this is mentioned first time in the end of section 2.1 at the end of the page 2. As there are other common choices for making inference for type of structured models discussed in the paper, I would suggest to mention already in the abstract and in the introduction that the ACV discussed in the paper is specifically for the marginal maximum likelihood estimation or other well defined optimization--feel free to specify what are the requirements--and not for, e.g., MCMC.

Relation to Prior Work: Yes.

Reproducibility: No

Additional Feedback: ### After rebuttal I have read the other reviews and the author rebuttal. My main concern was that the assumptions were not clearly stated. The authors have responded well. Both the recap (lines 1-9 in the rebuttal) and lines 31-41 are exactly what I wanted to see and these are presented in such compact form that I'm confident that the authors are able to include them in the revision. Due to the excellent rebuttal I'm increasing my score to 8.


Review 2

Summary and Contributions: The paper proposes a methodological extension of the infinitesimal jackknife (IJ) approximate cross-validation to data that are modeled via hidden Markov models or conditional random fields. This is done by applying an HMM or CRF likelihood to the general ACV objective function and algorithm. Theory for finite sample error bounds and algorithmic computational performance are shown. Several real-data examples are presented comparing their ACV method to exact CV; it is shown that the two are comparable in terms of loss, while ACV only has a small fraction of the runtime compared to exact CV.

Strengths: The claims in the paper seem to be correct. They appear to be fairly direct extensions of methodology and theory from previous generalized cross validation literature. The empirical evaluations presented appear to be interpreted correctly as well. The paper is relevant to the NeurIPS community. The method introduced in this paper could be used for any machine learning method involving latent variable estimation and cross-validation for hyperparameter selection.

Weaknesses: The major limitation of this paper is the empirical studies section. A greatly expanded experimental simulation section could be useful for studying the method under controlled or known conditions and for comparing the approach to other methods in the literature. The paper does not compare the performance of their ACV method to cited previous ACV methods, such as the naive Newton step (NS) or IJ methods explicitly mentioned in the paper. The novelty of the work is relatively minor. Its main contribution is the application of a likelihood from a family of models that is fairly well-known to a general framework for ACV from a previous paper. There are some new theoretical results with respect to the error bounds of IJ ACV specifically with respect to HMMs and CRFs. Some of the specific contributions of the work presented are somewhat oversold. The authors discuss the societal impact of their method in very broad terms; it may be more pertinent to consider specific applications or fields where ACV may be necessary for data with dependency structures. The authors also introduce their method as widely applicable to data with dependent observations, but their specific studies only apply to methods that model these dependencies via latent processes.

Correctness: Assuming that the cited previous literature on generalized cross-validation and approximate cross-validation are correct, the claims and methods here should be correct as well.

Clarity: The mathematical notation is understandable and fairly standard. The proofs are well-formatted Grammatical errors appear in the paper, but they do not severely detract from the understanding of the main ideas of the paper.

Relation to Prior Work: The authors clearly discuss how their ACV method extends work from previous ACV methods by applying it to models that don't assume i.i.d. data.

Reproducibility: Yes

Additional Feedback: There are a few minor grammatical errors throughout the paper. Figures in the real-data experiment section could be better labeled/presented, e.g. error as a percentage of original model MSE. The ACV vs. exact CV runtime graphs could either be bar graphs or just plotted like a normal line graph. Figure 2 placement needs to be adjusted in section 4, as the first line of a paragraph is split off. The paper refers to two different methods of leaving out data for the HMM model as "case A" and case "B"; these could be better named as currently it can be difficult for the reader to remember which case is which. The terminology in Proposition 1 is a little difficult to understand.


Review 3

Summary and Contributions: This paper proposes algorithms for approximate cross validation for structured data by developing infinitesimal jackknife approximations to circumvent an exact initial fit which is considered intractable in general for structured data. The paper provides some theoretical analysis on the approximation error. The paper empirically verifies the effectiveness of the proposed methods on multiple settings.

Strengths: 1) The problem of extending approximate cross validation to structured data is very important. 2) The paper provides both theoretical and empirical justifications.

Weaknesses: 1) The presentation of the paper can be substantially improved. For example, frequent switching between LSCV and LWCV can confuse readers. 2) In the empirical evaluation, it lacks quantitative results, which makes it hard to assess the performance of the proposed algorithms.

Correctness: looks correct to me

Clarity: okay but has room to improve.

Relation to Prior Work: yes

Reproducibility: Yes

Additional Feedback: 1) The authors are suggested to put key algorithms (e.g., Alg. 2) in the main text to make it self-contained. 2) Why do the authors choose 500 folds in the experiment? It sounds too large to use in practice. 3) Line 11-12: it reads that the authors have a solution for (ii), but it is not clear to me what the solution is.

[Author Response · NeurIPS 2020]

We thank the reviewers for their helpful comments. First, we briefly recap our contributions. We provide the first
computationally efficient and accurate method for approximate cross-validation (ACV) in the following setting:
structured models fit with MLE or MAP, where the task is a structured prediction. In so doing, we show (theoretically
and empirically) that IJ-style ACV ideas can be applied at an inexact (MLE or MAP) optimum; the latter is a new
result even for traditional ACV. By contrast, previous IJ approximations (e.g. Giordano et al. [2019]) do not apply to
structured prediction and also assume an exact optimum; Newton-step (NS) methods (e.g. Rad and Maleki [2020]) *could*
be applied to structured prediction, but we show (lines 118–121, Appendix G) they are not computationally efficient;
and Bürkner et al. [2020] focus on leave-future-out for time series (but not other forms of structured prediction) within
a fully Bayesian (rather than MLE or MAP) framework.

**Novelty**: (A) R2 is concerned that we are applying a "general framework for ACV from a previous paper." But note that
Giordano et al. [2019], Beirami et al. [2017], Koh & Liang [2017], Koh et al. [2019], Stephenson & Broderick [2020],
and Wilson et al. [2020] all work in the framework of Eq. (1) of Giordano et al. [2019], whose additive form does not
allow for the structured tasks we consider here. Observe that Bürkner et al. [2020] extend Bayesian ACV methods to
leave-future-out CV for time series models; that work demonstrates the non-trivial nature of extensions to structured
tasks. (B) R2 also describes our new theoretical error bounds as applying to HMMs and CRFs. But note that our error
bound (Prop. 2) applies much more broadly — both to all of the structured tasks we consider here, including more
general MRFs — as well the Giordano et al. [2019] framework with inexact optima (a result which was not previously
established but is more practically relevant than results requiring exact optima). (C) R2 is concerned that our methods
apply only to models with latent processes. While we believe models with latent structure represent a widely used and
interesting class of models, we note that we do present methodology (Sections E, F, and Algorithm 4) and experiments
(Section 5, lines 265–291) for CRFs, which contain no latent variables. And our inexact optimum theory applies beyond
structured models.

**Additional experiments**: R2 asks for comparisons against pre-existing IJ or NS ACV methods. Unfortunately, there
are no existing IJ methods that apply to the tasks we consider. To the best of our knowledge, NS has not previously
been applied to these problems, but we do consider it. In lines 118–121 and Appendix G, we discuss the computational
challenges of NS methods, which arise from computing and inverting a new Hessian for each CV fold. E.g., for the
time-varying Poisson process problem ($T = 50,400$), IJ computation across all $1,000$ folds took about 12 minutes. By
contrast, the NS approximation here would take roughly *113 hours*. On sufficiently small datasets, though, the NS and
our IJ can be much closer in performance. We will include NS timing across datasets, and discussion, in a revision of
our main text.

**Assumptions**: We fully agree with R1 that our assumptions could be stated more clearly. We will collect them in the
text, and we note them here for clarity. We require (1) a model fit via optimization, (2) twice differentiability of the
model objective and invertibility of the Hessian matrix at the initial model fit $\hat{\Theta}$, and (3) the ability to write the model
fits across CV folds, $\hat{\Theta}^{\backslash o}$, as optima of the same weighted objective for all folds $\mathbf{o}$. These conditions are satisfied
by a broad class of empirical/regularized risk minimization problems, including widely used probabilistic models of
structured data fit via MLE or MAP. We further clarify that *any* optimization method (stochastic or exact; with or
without early stopping; etc.) may be used to obtain the initial fit $\hat{\Theta}$ for input to Algorithm 1. When computing the
actual derivatives in Algorithm 1, we assume that the gradients are computed exactly at $\hat{\Theta}$. Computing these gradients
involves a single pass through the dataset before approximating CV for all folds.

**Structured tasks**: R1 notes that we need to clarify that our novelty here is for the combination of structured models
*paired with* structured tasks. We completely agree and will be sure to make this point very early in a revised manuscript.

**Number of folds**: R4 is concerned that 500 folds in our neural CRF experiment is "too large to use in practice." Fig. 1
of Rad and Maleki [2020] show leave-one-out CV (with hundreds of folds) substantially outperforming $\{3, 5, 10\}$-fold
CV at estimating out-of-sample error. A goal of modern ACV methods is to allow a larger number of folds, and
therefore more accurate final estimate overall, in practice.

**Quantitative results**: R4 is concerned that our paper "lacks quantitative results." We interpret this concern to mean
that the reviewer would like to see approximation error and timing reported in numbers, in addition to being plotted in
the figures we have included in our submission. We will be sure to include both numbers and figures in a revision.

**Inexact initial fit**: R4 is would like us to clarify how our method provides a solution to the inexact initial fit problem.
We prove (Section 4) that the IJ approximation error increases smoothly with the error in the initial fit. Proposition 2
explicitly bounds the approximation error and suggests that we can use "good enough" initial fits without substantively
sacrificing ACV accuracy. Our neural CRF experiments in Section 5 provide empirical confirmation.

*References* (beyond original paper) ∘ P. W. Koh, K. S. Ang, H. Teo, P. Liang. NeurIPS'19.
∘ A. Wilson, M. Kasy, L. Mackey. AISTATS'20.


[Meta-Review · NeurIPS 2020]

The reviewers agree that this submission represents an important contribution to the field. Please be sure to carefully review and address the concerns of all reviewers in the revision.